# Enhanced ER proteostasis and temperature differentially impact the mutational tolerance of influenza hemagglutinin

Angela M Phillips[1], Michael B Doud[2,3], Luna O Gonzalez[4], Vincent L Butty[5], Yu-Shan Lin[6], Jesse D Bloom[2,3], Matthew D Shoulders[1]*

[1]Department of Chemistry, Massachusetts Institute of Technology, Cambridge, United States; [2]Fred Hutchinson Cancer Research Center, Seattle, United States; [3]Department of Genome Sciences, University of Washington, Seattle, United States; [4]Department of Mathematics, Massachusetts Institute of Technology, Cambridge, United States; [5]BioMicro Center, Massachusetts Institute of Technology, Cambridge, United States; [6]Department of Chemistry, Tufts University, Medford, United States

**Abstract** We systematically and quantitatively evaluate whether endoplasmic reticulum (ER) proteostasis factors impact the mutational tolerance of secretory pathway proteins. We focus on influenza hemaggluttinin (HA), a viral membrane protein that folds in the host's ER via a complex pathway. By integrating chemical methods to modulate ER proteostasis with deep mutational scanning to assess mutational tolerance, we discover that upregulation of ER proteostasis factors broadly enhances HA mutational tolerance across diverse structural elements. Remarkably, this proteostasis network-enhanced mutational tolerance occurs at the same sites where mutational tolerance is most reduced by propagation at fever-like temperature. These findings have important implications for influenza evolution, because influenza immune escape is contingent on HA possessing sufficient mutational tolerance to evade antibodies while maintaining the capacity to fold and function. More broadly, this work provides the first experimental evidence that ER proteostasis mechanisms define the mutational tolerance and, therefore, the evolution of secretory pathway proteins.
DOI: https://doi.org/10.7554/eLife.38795.001

*For correspondence: mshoulde@mit.edu

Competing interests: The authors declare that no competing interests exist.

## Introduction

Protein evolution is necessarily constrained by the inherent biophysical properties of polypeptide sequences (*Wylie and Shakhnovich, 2011*; *Wyganowski et al., 2013*; *Bloom and Glassman, 2009*; *DePristo et al., 2005*; *Tokuriki and Tawfik, 2009a*; *Gong et al., 2013*). Gene variants that encode proteins unable to properly fold are selected against (*Figure 1A*) (*Powers et al., 2009*), irrespective of any theoretically beneficial new function the encoded protein may otherwise have acquired (*Wylie and Shakhnovich, 2011*; *Wyganowski et al., 2013*; *Bloom and Glassman, 2009*; *DePristo et al., 2005*; *Tokuriki and Tawfik, 2009a*; *Gong et al., 2013*). Importantly, the consequences of amino acid substitutions for protein folding depend not just on inherent biophysical properties, but also on the environment in which the nascent chain attempts to fold. In cells, this environment is defined most importantly by the proteostasis network, which consists of a subcellular compartment-specific array of chaperones and quality control factors (*Balch et al., 2008*; *Wong et al., 2018*; *Hartl et al., 2011*; *Shoulders et al., 2013*). These proteostasis factors assist

**Figure 1.** The proteostasis boundary model for protein evolution. (A) The accessibility of protein evolutionary intermediates is defined by protein stability, folding rate, and misfolding rate. Protein variants inside the boundary are accessible (shaded in black), and those outside the boundary are inaccessible (shaded in red). Figure generated using modified Mathematica macro from Powers et al (*Powers et al., 2009*). (B) In a biophysically permissive environment, such as cells with elevated levels of proteostasis factors, the boundary may be shifted, making otherwise non-functional variants accessible.

DOI: https://doi.org/10.7554/eLife.38795.002

client proteins in navigating complex folding landscapes (*Wyganowski et al., 2013*; *Tokuriki and Tawfik, 2009b*), and are thus theoretically well-positioned to modulate the mutational landscape accessible to evolving proteins (*Figure 1B*).

The proteostasis network consists of ~1000 factors involved in client protein folding, modification, trafficking, and quality control. Yet, to date, virtually all studies experimentally examining evolutionary consequences of eukaryotic proteostasis network factors have focused on a single chaperone – HSP90 (*Geller et al., 2012*; *Rohner et al., 2013*; *Queitsch et al., 2002*; *Cowen and Lindquist, 2005*; *Geller et al., 2018*; *Sangster et al., 2008*; *Geiler-Samerotte et al., 2016*). These focused studies have shown that HSP90 can (1) buffer the phenotypic effects of standing genetic diversity (*Rohner et al., 2013*; *Queitsch et al., 2002*; *Sangster et al., 2008*; *Geiler-Samerotte et al., 2016*) and (2) potentiate the phenotypic effects of new mutations (*Cowen and Lindquist, 2005*; *Geller et al., 2018*; *Geiler-Samerotte et al., 2016*), perhaps by facilitating protein folding and minimizing aggregation (*Geller et al., 2018*). Looking beyond HSP90, we recently leveraged chemical biology tools to demonstrate that other cytosolic proteostasis factors can also modulate protein evolutionary trajectories and mutational tolerance (*Phillips et al., 2017*).

The potentially critical impact of the extensive endoplasmic reticulum (ER) proteostasis machinery on the mutational tolerance and evolution of secretory pathway client proteins has remained unexplored. This gap in knowledge is important because roughly 1/3 of the proteome folds in the ER (*Wong et al., 2018*). Moreover, thousands of mutations in ER client proteins cause hundreds of currently incurable diseases, ranging from neurodegenerative pathologies to the lysosomal storage disorders and the collagenopathies (*Balch et al., 2008*). Additionally, viral membrane proteins often fold in the ER and must rapidly evolve to escape host immune factors (*Chen et al., 1995*; *Daniels et al., 2003*; *Doud et al., 2017*). Thus, understanding how ER chaperones influence the ability of client proteins to tolerate mutations and adapt to changing environments is essential as we work to design effective therapies for these diseases.

Influenza hemagglutinin (HA) is a uniquely attractive model protein for systematically and quantitatively evaluating whether and how ER proteostasis mechanisms impact client protein mutational tolerance. The folding of HA is perhaps as well-delineated as for any other membrane protein, and HA interacts extensively with components of the host cell's ER proteostasis network (*Chen et al., 1995*; *Daniels et al., 2003*; *Sauter et al., 1992*; *Nakajima et al., 1986*; *Frabutt et al., 2018*; *Hurtley et al., 1989*; *Hebert et al., 1997*; *Ueda and Sugiura, 1984*; *Skehel and Wiley, 2000*; *Klein et al., 2018*; *Gamblin et al., 2004*; *Pankow et al., 2015*). Co-translationally, HA is heavily *N*-glycosylated and engages the ER's lectin chaperones, calnexin and calreticulin, which increase HA's folding efficiency and prevent misfolding (*Chen et al., 1995*; *Daniels et al., 2003*; *Hebert et al.,*

*1997*). During the folding process, HA also interacts with protein disulfide isomerases to form its six disulfide bonds, as well as with binding immunoglobulin protein (BiP) and glucose-regulated protein 94 (GRP94), which are the ER homologs of HSP70 and HSP90, respectively (*Daniels et al., 2003*; *Hurtley et al., 1989*). Together, these host ER chaperones and folding enzymes assist HA monomer folding and prevent premature trimerization, ensuring that HA reaches its functional conformation or otherwise is targeted for ER-associated degradation (*Balch et al., 2008*; *Shoulders et al., 2013*; *Frabutt et al., 2018*).

Amino acid substitutions that impair HA stability, folding, or trafficking reduce HA fitness (*Bloom and Glassman, 2009*; *Hebert et al., 1997*; *Klein et al., 2018*). ER proteostasis factors that assist HA folding and trafficking are thus poised to modulate HA mutational tolerance (*Figure 1B*). Because HA variant fitness is directly coupled to viral replication via HA's role in host cell binding and fusion (*Sauter et al., 1992*; *Skehel and Wiley, 2000*), we believed that we could apply a deep mutational scanning strategy (*Fowler and Fields, 2014*) to quantitatively evaluate how ER proteostasis impacts the fitness of all viable single amino acid HA variants. Specifically, we aimed to quantify HA variant fitness by sequencing HA mutant viral libraries after selection in modified ER proteostasis environments, an approach that is substantially more sensitive and higher-throughput than the alternative functional assays employed for deep mutational scanning of non-viral secretory pathway clients (*Fowler and Fields, 2014*).

In summary, HA is an attractive model secretory protein because HA folding is well-characterized, HA interacts extensively with host ER proteostasis factors, and HA mutational tolerance can be comprehensively quantified by pairing deep mutational scanning with sequencing. Moreover, understanding factors that both constrain and enhance HA mutational tolerance has direct therapeutic relevance. HA is the primary target of neutralizing antibodies (*Doud et al., 2017*), and influenza immune escape is thus contingent on HA possessing sufficient mutational tolerance to acquire antibody resistance while still maintaining the capacity to fold and function. The potent impact of host chaperones on viral evolutionary trajectories has only recently been realized (*Phillips et al., 2017*), and it remains unknown whether, and to what degree, host proteostasis mechanisms impact viral protein mutational tolerance.

Below, we integrate deep mutational scanning with small molecule control of ER proteostasis environments to test the hypothesis that upregulating host ER proteostasis factors mitigates the fitness consequences of otherwise biophysically deleterious amino acid substitutions in HA (*Figure 1B*). We examine HA mutational tolerance in cells with basal versus upregulated levels of ER proteostasis factors, both at normal body temperature and at an elevated, fever-like temperature. We find that upregulation of the ER proteostasis machinery broadly enhances HA mutational tolerance, while increased temperature generally reduces HA mutational tolerance. Remarkably, the same HA sites where variants are most commonly temperature-sensitive, and therefore most likely to be biophysically problematic, are also the sites where variants benefit most from upregulation of the host's ER proteostasis machinery. Thus, enhanced host ER proteostasis mechanisms and increased temperature modulate HA mutational tolerance in opposite directions, a finding with important implications for influenza evolution. More broadly, this work provides the first experimental evidence that the composition of the ER proteostasis network can profoundly impact the mutational tolerance and evolution of secretory pathway clients.

## Results

### Modulating ER proteostasis during influenza infection

The composition of the ER proteostasis network is regulated by a stress-responsive signaling pathway termed the unfolded protein response (UPR). Under ER stress conditions, the UPR is activated and the IRE1 and ATF6 signaling cascades upregulate distinct but overlapping sets of ER proteostasis network components via the XBP1s and ATF6f transcription factors, respectively (*Shoulders et al., 2013*). Here, we profiled HA mutational tolerance at both a normal (37°C) and a fever-like (39°C) temperature in distinctive host ER proteostasis environments. ER proteostasis was modulated either by inducing XBP1s alone or by inducing XBP1s in concert with ATF6f in a small molecule-mediated, stress-independent manner in HEK293[ATF6f/XBP1s] cells using previously characterized chemical genetic methods (*Shoulders et al., 2013*). In these cells, activation of tetracycline

repressor-regulated XBP1s by the small molecule doxycycline induces the transcriptional response downstream of IRE1, remodeling the ER proteostasis environment by enhancing transcript levels of ~180 ER chaperones and quality control factors (*Wong et al., 2018*; *Shoulders et al., 2013*). Similarly, induction of destabilized domain-regulated ATF6f by the small molecule trimethoprim in HEK293[ATF6f/XBP1s] cells increases transcript expression of ~40 ER chaperones and quality control factors that are distinct from but partially overlapping with those regulated by XBP1s (*Wong et al., 2018*; *Shoulders et al., 2013*). Concomitant induction of ATF6f and XBP1s creates a third, distinct ER proteostasis environment, increasing expression of ~350 genes (*Shoulders et al., 2013*). We specifically assessed HA mutational tolerance in a basal ER proteostasis environment and in the two most significantly remodeled proteostasis environments, activating XBP1s alone or simultaneously activating XBP1s and ATF6f.

The media composition and treatment regimen for influenza propagation differed from the conditions used in the original characterization of XBP1s and ATF6f/XBP1s activation in these cells, which also was performed only at 37°C (*Shoulders et al., 2013*). Hence, before quantifying HA mutational tolerance in these distinctive cellular settings, we first comprehensively characterized each environment in the absence of an influenza infection using RNA-seq. We examined changes in ER and cytosolic proteostasis factor transcript expression (*Figure 2*), as well as global transcriptional changes to assess any off-target transcriptional effects (*Figure 2—figure supplement 1*). Essentially all transcripts upregulated in the original characterization of these cells (*Shoulders et al., 2013*) were also upregulated in response to our modified treatment regimen (*Figure 2—figure supplement 2*). Likely owing to the increased sensitivity of RNA-Seq compared to whole-genome arrays (*Marioni et al., 2008*), we also observed upregulation of 201 transcripts upon XBP1s induction and 268 transcripts upon ATF6f/XBP1s induction that were not observed to be significantly altered in the previous analysis. Still, as expected (*Shoulders et al., 2013*), transcripts upregulated upon XBP1s and ATF6f/XBP1s induction largely corresponded to secretory pathway and ER stress response genes (*Figure 2—figure supplement 1* and *Figure 2—source data 2*).

Specifically, we found that XBP1s and ATF6f/XBP1s activation in HEK293[ATF6f/XBP1s] cells selectively enhanced transcript expression of numerous ER proteostasis factors, including many known to interact with HA, such as calnexin (*CANX*) (*Chen et al., 1995*; *Daniels et al., 2003*; *Hebert et al., 1997*), calreticulin (*CALR*) (*Chen et al., 1995*; *Daniels et al., 2003*; *Hebert et al., 1997*), and members of the protein disulfide isomerase family (*PDI*) (*Figure 2A*, top) (*Daniels et al., 2003*). Concomitant induction of XBP1s and ATF6f revealed increased expression of ATF6f and ATF6f/XBP1s targets, including BiP (*HSPA5*) and GRP94 (*HSP90B1*), respectively (*Figure 2A*, top). Stress-independent activation of XBP1s or ATF6f/XBP1s did not significantly impact transcript levels of cytosolic protein folding factors at either 37°C or 39°C, highlighting the selectivity of this approach for modulating ER proteostasis (*Figure 2A–B*, bottom; *Figure 2—figure supplement 1A–D*). Increasing the host cell temperature to 39°C induced significantly fewer transcriptional changes than did activating XBP1s or ATF6f/XBP1s. Increased temperature did result in mild induction of the heat shock response, as expected (illustrated by elevated HSP70 (*HSPA1A*) levels in *Figure 2C* and *Figure 2—figure supplement 1E*).

We next assessed whether induction of the host cell's ER proteostasis network and increased temperature altered host cell viability or prevented influenza propagation. First, we activated XBP1s alone or with ATF6f at both 37°C and 39°C in the absence of influenza, and evaluated cell viability 72 hr later, corresponding to the duration of our intended drug pre-treatment and influenza infection. Consistent with previous studies (*Shoulders et al., 2013*), these perturbations were not cytotoxic (*Figure 2—figure supplement 3A*). Next, we evaluated whether influenza could propagate in these remodeled host cells. Briefly, we infected cells in each pre-activated selection environment with wild-type influenza A/WSN/1933 (the strain used throughout this work; hereafter referred to as influenza) and titered the viral supernatant 48 hr post-infection using a TCID$_{50}$ assay. Influenza growth was moderately attenuated at increased temperature (*Figure 2—figure supplement 3B*), but was still sufficient for performing selections and evaluating HA mutational tolerance by sequencing.

Finally, we evaluated whether our methods for perturbing ER proteostasis were functional during an influenza infection cycle. Using qPCR, we found that XBP1s and ATF6f target genes were still inducible by our chemical genetic methods in the presence of influenza, with only slightly attenuated upregulation relative to a mock-infection (*Figure 2—figure supplement 3C*). This modest reduction in chaperone transcript expression was likely caused by global suppression of host transcription by

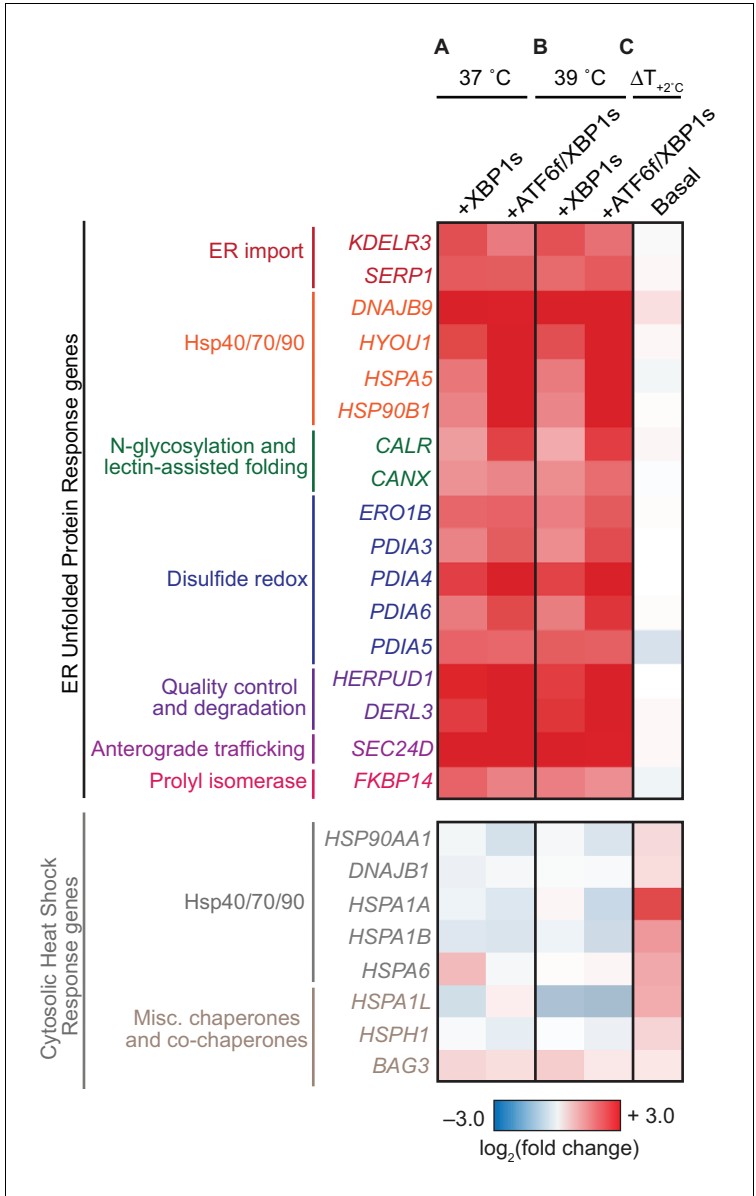

**Figure 2.** Induction of XBP1s and ATF6f/XBP1s is selective and does not globally induce stress responses. (**A**) Heat map of selected transcript-level effects of upregulating ER folding machinery in HEK293[ATF6f/XBP1s] cells via XBP1s or ATF6f/XBP1s at 37°C, relative to a basal environment at 37°C. (**B**) Heat map of selected transcript-level effects of upregulating ER proteostasis machinery via XBP1s or ATF6f/XBP1s at 39°C, relative to a basal environment at 39°C. (**C**) Heat map of selected transcript-level effects of increasing the temperature from 37°C to 39°C in a basal ER proteostasis environment for 24 hr. Genes in **A–C** grouped by ER (top) or cytosolic (bottom) proteostasis networks, and also based on functional classifications (*Shoulders et al., 2013*). For (**A–B**), tetracycline repressor-regulated XBP1s was induced by treatment with 0.1 μg/mL doxycycline (24 hr); destabilized domain-regulated ATF6f was induced by treatment with 1 μM trimethoprim (24 hr). *Figure 2—figure supplement 1*. Full transcriptome analysis confirmed selectivity of XBP1s and ATF6f/XBP1s induction. *Figure 2—figure supplement 2*. Comparison of HEK[ATF6f/XBP1s] RNA-Seq characterization to previous whole-genome array characterization. *Figure 2—figure supplement 3*. Methods to induce XBP1s and ATF6f/XBP1s did not cause cytotoxicity and were functional during influenza infection. *Figure 2—source data 1*. Complete RNAseq differential expression analysis. *Figure 2—source data 2*. Comparison of HEK[ATF6f/XBP1s] characterization to previous characterization. *Figure 2—source data 3*. Fold-change of HSR and UPR targets during influenza infection.
DOI: https://doi.org/10.7554/eLife.38795.003

The following source data and figure supplements are available for figure 2:

**Source data 1.** Complete RNAseq differential expression analysis.

*Figure 2 continued on next page*

*Figure 2 continued*

DOI: https://doi.org/10.7554/eLife.38795.007

**Source data 2.** Comparison of HEK^ATF6f/XBP1s characterization to previous characterization.

DOI: https://doi.org/10.7554/eLife.38795.008

**Source data 3.** Fold change of HSR and UPR targets during influenza infection.

DOI: https://doi.org/10.7554/eLife.38795.009

**Figure supplement 1.** Full transcriptome analysis confirmed selectivity of XBP1s and ATF6f/XBP1s induction.

DOI: https://doi.org/10.7554/eLife.38795.004

**Figure supplement 2.** Comparison of HEK^ATF6f/XBP1s RNA-Seq characterization to previous whole-genome array characterization.

DOI: https://doi.org/10.7554/eLife.38795.005

**Figure supplement 3.** Methods to induce XBP1s and ATF6f/XBP1s are not cytotoxic and are functional during influenza infection.

DOI: https://doi.org/10.7554/eLife.38795.006

influenza (*Russell et al., 2018*). However, overall our tools for modulating ER proteostasis factors maintained efficacy during the course of an influenza infection.

## Deep mutational scanning of HA in modulated ER proteostasis environments

To systematically profile HA mutational tolerance in modulated ER proteostasis environments, we employed deep mutational scanning (DMS), a method that couples saturating mutagenesis with fitness measurements, thereby efficiently sampling amino acid sequence space (*Doud et al., 2017*; *Fowler and Fields, 2014*; *Doud and Bloom, 2016*). We used biological triplicate HA viral mutant libraries that were previously generated from independent HA plasmid libraries using cells with basal levels of proteostasis factors at 37°C (*Doud and Bloom, 2016*). The resulting HA mutant viral libraries contained nearly all viable single amino acid HA variants (*Doud and Bloom, 2016*). Variant sensitivity to ER proteostasis mechanisms was tested by simultaneously competing variants within each viral library in host cells with modulated ER proteostasis environments. Each of these competitions was carried out at both 37°C and 39°C, to impose additional biophysical selection pressure on HA and to mimic elevated host body temperature during an influenza infection (*Figure 3A*).

We designed these DMS experiments to fully sample the library diversity whilst minimizing co-infection and genetic hitchhiking. To achieve this goal, we infected cells in each pre-activated selection environment with $10^6$ virions at an MOI of 0.01 infectious virion/cell. Following a 48 hr infection, or about four replication cycles, we harvested viral RNA and prepared subamplicon sequencing libraries for accurate variant frequency quantitation (*Doud and Bloom, 2016*). We then quantified the differential selection (diffsel) for each mutation as the logarithm of its enrichment in the selection condition relative to that of the mock-selection condition (e.g., Basal 39°C relative to Basal 37°C) (*Doud et al., 2017*; *Bloom, 2015*). Hence, the wild-type residue at each site has a diffsel of zero, variants with positive diffsel values are more fit in the selection condition than the mock condition relative to wild-type, and variants with negative diffsel values are less fit in the selection condition than the mock condition relative to wild-type. The diffsel for variants across HA can be visualized with sequence logo plots, where the size of the amino acid letter abbreviations is proportional to the magnitude of the differential selection for that variant (*Figure 3B*) (*Doud et al., 2017*).

To decipher selection from the inherent experimental noise of batch competitions, we filtered the DMS data to include only variants with robust differential selection across biological triplicates. Specifically, we limited our analyses to variants present in each pre-selection replicate viral library (*Figure 3C—Triplicate mutant libraries*) that exhibited differential selection in the same direction across biological triplicates. These criteria reduced experimental noise owing to differences in replicate library composition, as well as uneven sampling and biased PCR amplification of low frequency variants. This filtering predominantly excluded variants that were negligibly affected by the selection, which often had slightly positive diffsel values in one replicate versus slightly negative diffsel values in another replicate. These criteria also excluded strongly enriched or depleted variants that were not present in triplicate pre-selection libraries and, less frequently, strongly enriched or depleted variants that did not behave reproducibly across triplicate selections. Notably, the unfiltered

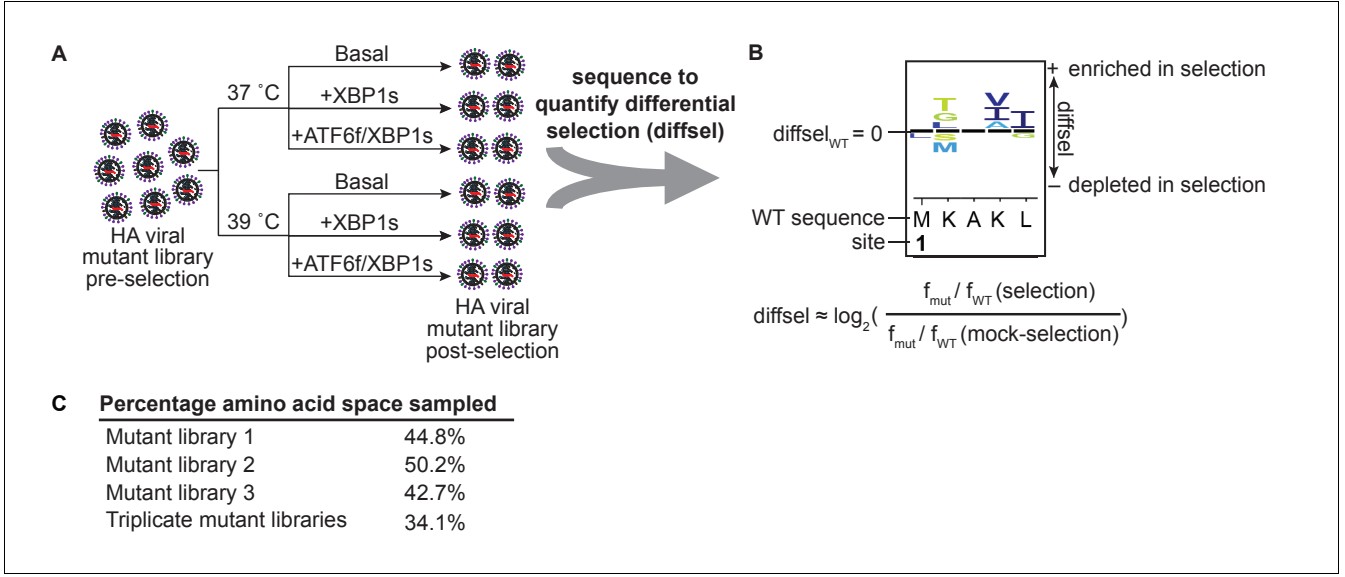

**Figure 3.** Deep mutational scanning of HA in modulated ER proteostasis environments. (A) Scheme for performing deep mutational scanning, and (B) quantifying the differential selection (diffsel) for each mutant in the selection environments, which can be visualized on sequence logo plots. In A, XBP1s or ATF6f/XBP1s were induced at either 37°C or 39°C for 16 hr prior to the selection, which consisted of a 48 hr infection. In B, $f_{WT}$ and $f_{mut}$ denote the wild-type and mutant frequencies, respectively. (C) Percentage of amino acid space sampled by each replicate library prior to filtering, and the percentage sampled by all three replicate mutant libraries. *Figure 3—source data 1*. Correlation coefficients for unfiltered absolute site differential selection values between biological replicate selections.

DOI: https://doi.org/10.7554/eLife.38795.010

The following source data is available for figure 3:

**Source data 1.** Correlation coefficients for unfiltered absolute site differential selection values between biological replicate selections.
DOI: https://doi.org/10.7554/eLife.38795.011

biological replicate diffsel values were strongly correlated across HA sites, with correlation coefficients ranging from $R$ = 0.67–0.78 (*Figure 3—source data 1* and *Source data 1*).

The filtered DMS data revealed that many HA variants were substantially depleted upon increasing the temperature (*Figure 4A*; *Figure 4—figure supplement 1*). In contrast, many variants were moderately enriched upon induction of ER proteostasis machinery (*Figure 4B–E*; *Figure 4—figure supplements 2–5*). This result was consistent with our hypothesis, as we anticipated that increasing the temperature would create a biophysically restrictive environment, reducing the fitness of HA variants, whereas increasing levels of ER proteostasis factors would create a biophysically permissive environment, enhancing the fitness of HA variants (*Figure 1B*).

To globally examine the relative enrichment of HA variants, we plotted HA mutant diffsel values and examined the deviation of the distribution mean from zero, which indicates no differential selection (*Figure 5A*). This analysis confirmed that increased temperature indeed generally reduced mutant HA viral growth, resulting in depletion (i.e., diffsel < 0) of 597 variants versus enrichment (i.e., diffsel > 0) of only 361 variants, relative to wild-type HA (*Figure 5A*). Variant depletion at elevated temperature could be explained by reduced variant folding rates and thus slower variant viral growth, or perhaps by increased degradation. Although we do not observe upregulation of ER-associated degradation factors at elevated temperature (*Figure 2C* and *Figure 2—figure supplement 1E*), it is certainly possible that HA variants engage degradation factors more at an elevated temperature.

In contrast, induction of XBP1s alone or with ATF6f generally enhanced mutant HA viral growth at both 37°C and 39°C, resulting in an overall enrichment of HA variants relative to wild-type HA (*Figure 5A*). The enhanced variant growth upon XBP1s induction alone was quite similar to that of simultaneous induction of XBP1s and ATF6f (*Figure 5—figure supplement 1* and *Figure 5—source data 1*), and is thus either predominantly caused by XBP1s, or by XBP1s-regulated factors that are redundant with those regulated by ATF6f. In either case, variant enrichment upon induction of

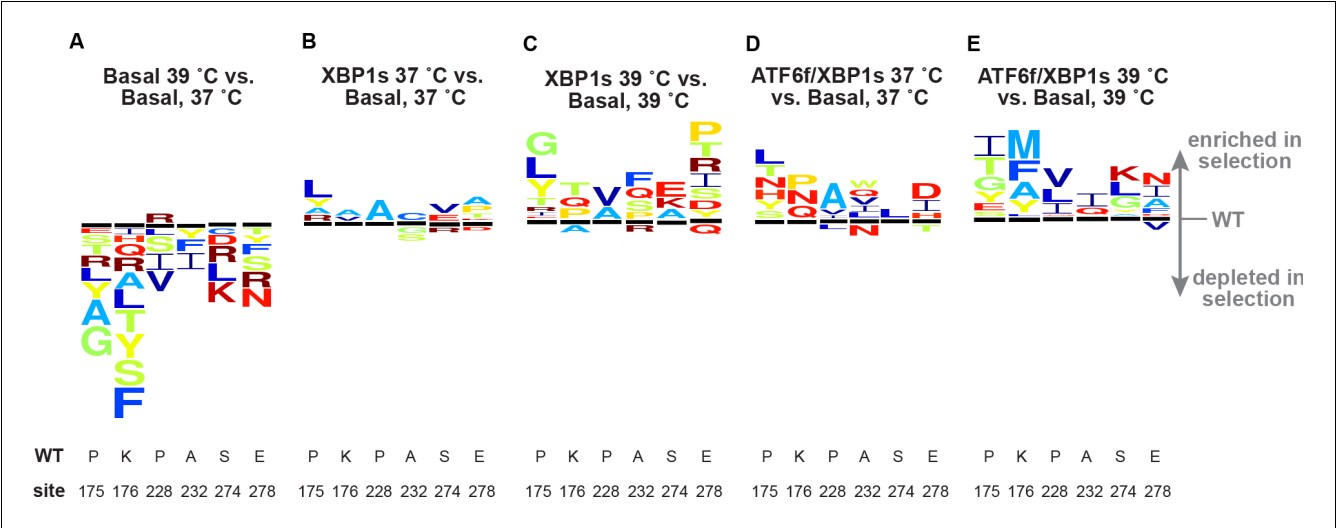

**Figure 4.** HA variants were depleted at increased temperature but enriched upon XBP1s or ATF6f/XBP1s induction. Cropped sequence logo plots show differential selection on HA at positions where variants were depleted upon increased temperature but enriched upon XBP1s and ATF6f/XBP1s induction (based on sequential numbering of WSN HA). Size of amino acid abbreviation corresponds to magnitude of selection. Only variants behaving consistently across biological triplicates are plotted; all selections are plotted on the same scale. *Figure 4—figure supplement 1*. Full WSN HA sequence logo plot: Basal 39°C vs. Basal 37°C. *Figure 4—figure supplement 2*. Full WSN HA sequence logo plot: XBP1s 37°C vs. Basal 37°C. *Figure 4—figure supplement 3*. Full WSN HA sequence logo plot: XBP1s 39°C vs. Basal 39°C. *Figure 4—figure supplement 4*. Full WSN HA sequence logo plot: ATF6f/XBP1s 37°C vs. Basal 37°C. *Figure 4—figure supplement 5*. Full WSN HA sequence logo plot: ATF6f/XBP1s 39°C vs. Basal 39°C. *Source data 1*. Complete analysis of deep mutational scanning data.
DOI: https://doi.org/10.7554/eLife.38795.012

The following figure supplements are available for figure 4:

**Figure supplement 1.** Full WSN HA sequence logo plot: Basal 39°C vs. Basal 37°C.
DOI: https://doi.org/10.7554/eLife.38795.013

**Figure supplement 2.** Full WSN HA sequence logo plot: XBP1s 37°C vs. Basal 37°C.
DOI: https://doi.org/10.7554/eLife.38795.014

**Figure supplement 3.** Full WSN HA sequence logo plot: XBP1s 39°C vs. Basal 39°C.
DOI: https://doi.org/10.7554/eLife.38795.015

**Figure supplement 4.** Full WSN HA sequence logo plot: ATF6f/XBP1s 37°C vs. Basal 37°C.
DOI: https://doi.org/10.7554/eLife.38795.016

**Figure supplement 5.** Full WSN HA sequence logo plot: ATF6f/XBP1s 39°C vs. Basal 39°C.
DOI: https://doi.org/10.7554/eLife.38795.017

XBP1s or ATF6f/XBP1s is likely explained by upregulation of ER protein folding factors (*Figure 2A–B*), while the lesser amount of variant depletion observed upon induction of XBP1s or ATF6f/XBP1s may be attributed to upregulation of ER quality control and degradation factors (*Figure 2A–B*). Importantly, we note that the enrichment and depletion of HA variants may also be caused by indirect effects of inducing XBP1s or ATF6f/XBP1s and increasing temperature. For example, these perturbations could potentially impact the protein levels and activities of other host proteins that engage HA, or perhaps differentially modulate viral growth kinetics.

Regardless of their precise origins, the generally opposing effects of increased temperature and upregulated proteostasis machinery on mutant HA viral growth motivated examination of these effects on individual HA variants. To this end, we determined the correlation between the average diffsel values for increased temperature and upregulated ER proteostasis machinery for individual HA variants (*Figure 5B–C*). This analysis revealed significant negative correlation between these selection conditions, demonstrating that the HA variants most depleted upon increased temperature corresponded to those most enriched upon induction of ER proteostasis factors.

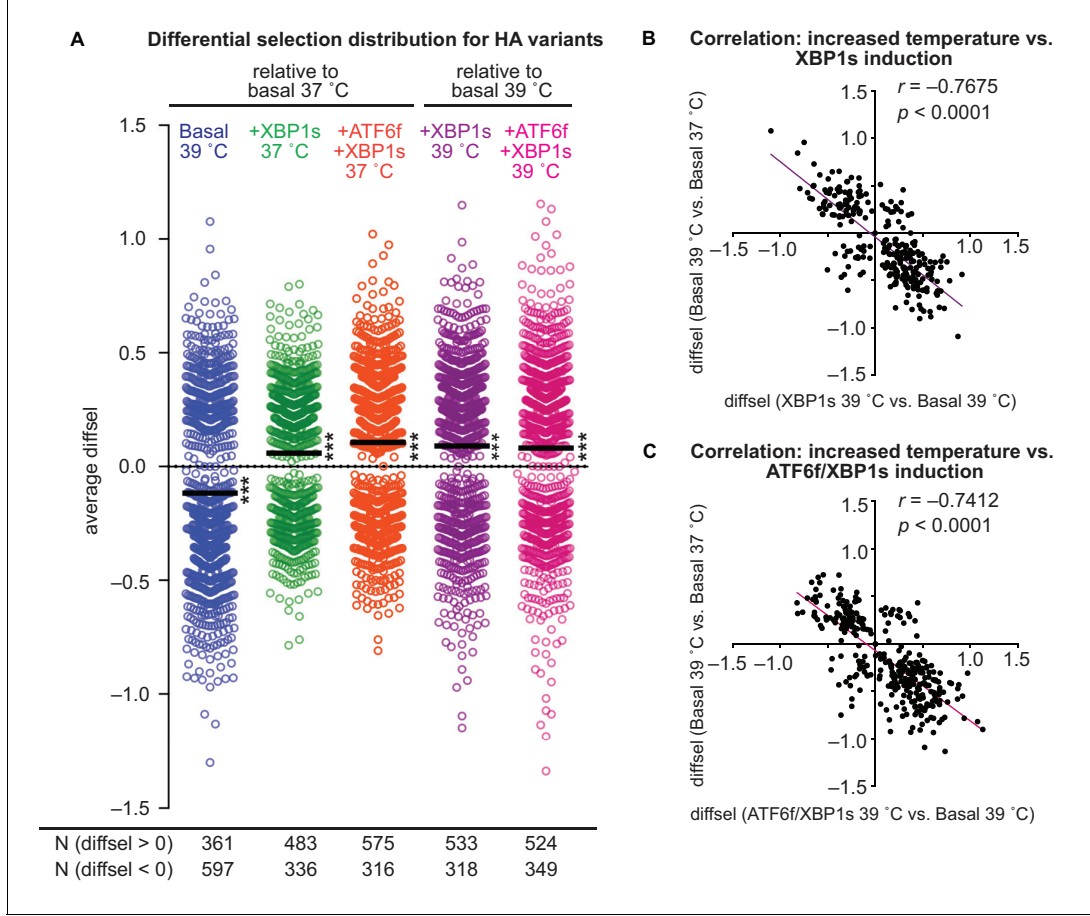

**Figure 5.** ER proteostasis mechanisms and temperature divergently impact HA variant viral growth. (**A**) Average differential selection (diffsel) for each HA variant; the black line designates the mean of the distribution. Significance of deviation of the mean from zero (no differential selection) was tested by a one-sample $t$-test; *** designates two-tailed $p$-values≤0.001. Variant diffsel values are staggered about the x-axis to minimize overlap. The number of variants with diffsel >0 and <0 is listed below each distribution. (**B**) Correlation for diffsel values for inducing XBP1s at 39°C versus increasing the temperature in basal cells (N = 271). (**C**) Correlation for diffsel values for inducing ATF6f/XBP1s at 39°C versus increasing the temperature in basal cells (N = 320). For (**A–C**), average diffsel values for variants behaving consistently across biological triplicates are plotted. For (**B–C**), Pearson correlation and $p$-value from an $F$-test are shown. *Figure 5—figure supplement 1*. Induction of ATF6f/XBP1s and XBP1s similarly impact HA variant viral growth. *Figure 5—source data 1*. Differential selection values for each selection, pre- and post-filtering.

DOI: https://doi.org/10.7554/eLife.38795.018

The following source data and figure supplement are available for figure 5:

**Source data 1.** Differential selection values for each selection, pre- and post-filtering.

DOI: https://doi.org/10.7554/eLife.38795.020

**Figure supplement 1.** Induction of ATF6f/XBP1s and XBP1s similarly impact HA variant viral growth.

DOI: https://doi.org/10.7554/eLife.38795.019

## Examination of variants and sites impacted most by host proteostasis and temperature

The opposing fitness effects of proteostasis mechanisms and temperature in the DMS data suggested that temperature-sensitive HA variants were most likely to be rescued by ER proteostasis factors. The potential implications of these opposing selection forces on HA motivated us to more rigorously evaluate this behavior for individual HA variants. DMS batch competitions are inherently noisy, largely owing to uneven sampling of library variants (*Doud and Bloom, 2016*). Hence, we performed pairwise competitions between wild-type influenza and influenza encoding differentially selected variants identified in the DMS. These competitions were performed in cells with either basal or elevated levels of ER proteostasis factors, accessed by inducing XBP1s alone or in combination

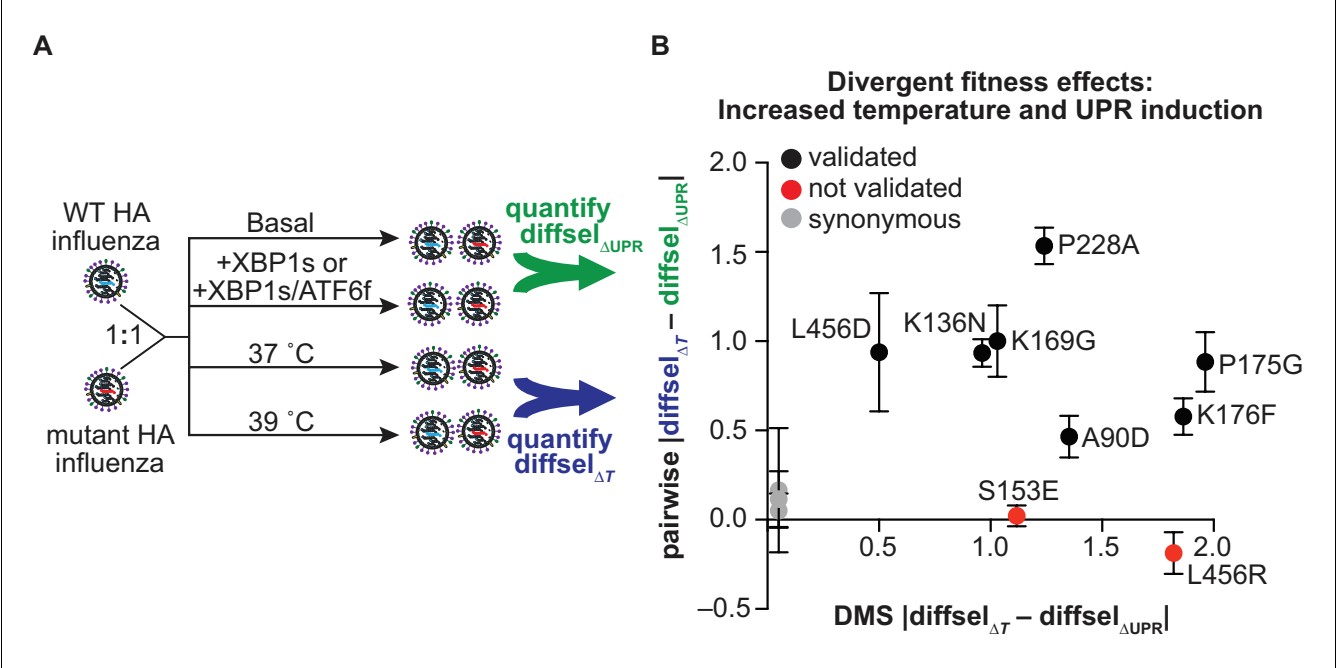

**Figure 6.** Divergent fitness effects revealed by DMS assessed by pairwise competitions. (**A**) Scheme for pairwise competition between wild-type HA and mutant HA influenza. Detailed description of competition conditions in *Figure 6—figure supplement 1*. (**B**) The difference between the average diffsel upon increased temperature (ΔT) and the average diffsel upon induction of XBP1s or ATF6f/XBP1s (ΔUPR) is plotted. This difference from the pairwise competitions is plotted on the y-axis, with error bars representing SEM for biological triplicates, and the corresponding difference from the DMS is plotted on the x-axis. Variants that validated (i.e., diffsel upon increased temperature was significantly different from that upon UPR induction in both DMS and pairwise competitions) are in black; variants that did not validate are in red; synonymous (neutral) validated negative controls are in gray. All non-synonymous variants are labeled. *Figure 6—figure supplement 1*. Individual pairwise competition diffsel values.

DOI: https://doi.org/10.7554/eLife.38795.021

The following figure supplement is available for figure 6:

**Figure supplement 1.** Individual pairwise competition diffsel values.

DOI: https://doi.org/10.7554/eLife.38795.022

with ATF6f, and at 37°C and 39°C (*Figure 6A*). We completed these pairwise competitions in biological triplicate for nine variants that in the DMS exhibited the most divergent behavior upon induction of XBP1s or ATF6f/XBP1s compared to increased temperature (A90D, K136N, S153E, K169G, P175G, K176F, P228A, L456D, and L456R; based on sequential numbering of WSN HA sites). We also included three synonymous negative controls (P175Psyn, P228Psyn, L456Lsyn), for which we mutated the wild-type codon to a synonymous codon that was not differentially selected in the DMS batch competition.

Comparison of the differential selection upon increased temperature with that upon XBP1s or ATF6f/XBP1s induction in the pairwise competitions recapitulated the divergent behavior observed in DMS for several individual HA variants. In particular, of the nine non-synonymous variants tested, seven variants exhibited significantly different behavior upon increased temperature versus XBP1s or ATF6f/XBP1s induction (*Figure 6B*—black; *Figure 6—figure supplement 1A*). This high rate of reproducibility lends substantial credence to the overall DMS data set. We note that some discrepancy between the DMS and pairwise competitions (*Figure 6B*—red; *Figure 6—figure supplement 1B*) is expected given the difference in wild-type frequencies between the DMS and pairwise competitions and the inherent false discovery rate of high-throughput screens (*Doud and Bloom, 2016*). As in the DMS (*Figure 4*), the pairwise diffsel upon increased temperature was overall more substantial than that upon XBP1s or ATF6f/XBP1s induction, though XBP1s or ATF6f/XBP1s induction did significantly impact the growth of K136N, P228A, and L456D (*Figure 6—figure supplement 1A*—see asterisks above individual bars). Importantly, all of the synonymous variants displayed similarly neutral behavior in the pairwise and DMS competitions (*Figure 6B*—gray; *Figure 6—figure*

*supplement 1C*), indicating that the opposing nature of these selection pressures is unlikely to reflect experimental noise.

Validation of the opposing effects of temperature and host ER proteostasis mechanisms on HA variant growth in the pairwise competitions prompted us to further examine the properties of HA sites most impacted by these selection environments. For HA variants present in each replicate mutant library (*Figure 3C*—Mutant libraries 1–3), we summed the mutation differential selection values at each site to calculate the net site differential selection (net site diffsel), a measure of site mutational tolerance, and averaged the net site diffsel values across the triplicate libraries (*Source data 1*). Consistent with the mutation diffsel values in *Figure 5A*, the net site diffsel values were negatively skewed upon increased temperature, indicating reduced mutational tolerance, but positively

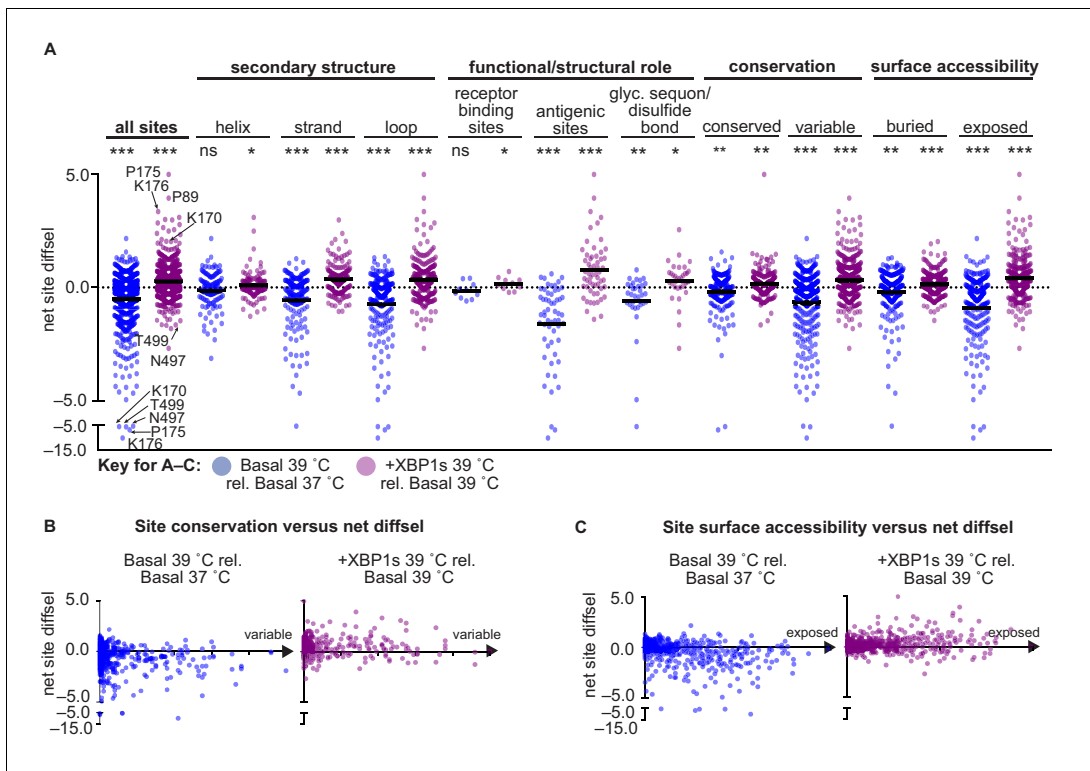

**Figure 7.** XBP1s induction and temperature impacted HA mutational tolerance at sites across HA with diverse functional and structural roles. (A) Average net site diffsel (sum of mutation diffsel values at a given site) is plotted for increased temperature in a basal environment (blue) and upon induction of XBP1s at 39°C (purple). Sites are sorted by secondary structure, functional/structural role, conservation in H1 sequences, and surface accessibility (buried residues are the 50% least surface-accessible sites). Select outliers are labeled. Means are represented by black lines and significance of deviation from zero (no selection) was determined by a one-sample *t*-test and is indicated above each distribution; *, **, and *** represent one-tailed p-values≤0.05, 0.01, and 0.001, respectively; ns corresponds to a p-value>0.05. (B) Average net site diffsel is plotted as a function of site conservation, where x = 0 corresponds to a completely conserved site among 989 non-redundant human H1 sequences (1918–2008) (*Zhang et al., 2017*). (C) Average net site diffsel is plotted as a function of site surface accessibility, where x = 0 corresponds to a buried residue (PDBID 1RVX [*Gamblin et al., 2004*]). *Figure 7—figure supplement 1*. ATF6f/ XBP1s induction and temperature impacted HA mutational tolerance at sites across HA with diverse functional and structural roles. *Figure 7—source data 1*. Properties and differential selection for each site of HA.
DOI: https://doi.org/10.7554/eLife.38795.023

The following source data and figure supplement are available for figure 7:

**Source data 1.** Properties and differential selection for each site of HA.
DOI: https://doi.org/10.7554/eLife.38795.025

**Figure supplement 1.** ATF6f/XBP1s induction and temperature impacted HA mutational tolerance at sites across HA with diverse functional and structural roles.
DOI: https://doi.org/10.7554/eLife.38795.024

skewed upon induction of XBP1s, indicating enhanced mutational tolerance (*Figure 7A*). Next, we sorted sites based on secondary structure to determine whether sites most affected by temperature or XBP1s induction belonged to particular secondary structural elements. We found that sites distributed throughout all secondary structural elements had enhanced mutational tolerance upon XBP1s induction, and that sites in strands and loops had particularly reduced mutational tolerance upon increased temperature. Thus, these selection pressures broadly impact HA mutational tolerance (*Figure 7A*; secondary structure).

We next plotted the net site diffsel values for sites essential for HA function and structure, to determine if these sites were impacted by temperature or ER proteostasis factors. Receptor binding sites had slightly, but significantly, enhanced mutational tolerance upon XBP1s induction, and were unaffected by increased temperature. In contrast, the mutational tolerance at antigenic sites was substantially impacted by both XBP1s induction and increased temperature (*Figure 7A*; functional/structural role). Moreover, sites in N-glycosylation sequons and cysteine residues involved in disulfide bonding, which engage glycoprotein chaperones and protein disulfide isomerases, respectively (*Chen et al., 1995*; *Daniels et al., 2003*; *Hebert et al., 1997*), had enhanced mutational tolerance upon XBP1s induction and reduced mutational tolerance upon increased temperature (*Figure 7A*; glyc. sequon/disulfide bond). Some of these sites—discussed in more detail below—were among the most affected by temperature and ER proteostasis factors.

Furthermore, we assessed whether ER proteostasis mechanisms and temperature differentially impact HA mutational tolerance at sites based on evolutionary conservation and surface accessibility. Specifically, we evaluated site conservation across 989 human H1 sequences from the Influenza Research Database (*Zhang et al., 2017*). We considered completely conserved sites as conserved and all other sites as variable, and plotted the net site diffsel values for conserved and variable sites (*Figure 7A*; conservation), as well as the site conservation as a function of the net site diffsel (*Figure 7B*). This analysis illustrated that temperature and XBP1s induction affected mutational tolerance at both variable and conserved sites (*Figure 7A*; conservation). However, the mutational tolerance at variable sites was more affected than at conserved sites, as indicated by the net site diffsel values for conserved sites clustering near zero, whereas the net site diffsel values for variable sites were more evenly distributed along the y-axis (*Figure 7B*). Still, the modest impact on conserved sites suggested that host temperature and proteostasis mechanisms could modulate mutational tolerance at essentially any site in HA, regardless of the operating structural and functional constraints. This conclusion is consistent with the distribution of net site diffsel values as a function of surface accessibility, which revealed that mutational tolerance was impacted by XBP1s induction and temperature at both buried and surface-exposed sites (*Figure 7A*; surface accessibility), though surface-exposed sites were often more affected (*Figure 7C*). We note that, similar to the mutation diffsel distributions (*Figure 5A*), the site diffsel distributions for ATF6f/XBP1s induction mirrored those for XBP1s induction (*Figure 7—figure supplement 1*).

Finally, we examined the structural location of HA sites most impacted by host proteostasis mechanisms and temperature. Mapping net site diffsel values onto the HA crystal structure revealed that these selection pressures affect the mutational tolerance of sites across HA (*Figure 8*; *Figure 8—figure supplement 1* (ATF6f/XBP1s)). Many HA sites strongly impacted by host ER proteostasis factors and temperature were in the globular head domain (*Figure 8A*). For example, K170, P175, and K176 were among the most mutationally intolerant sites upon increased temperature and the most mutationally tolerant sites upon induction of XBP1s or ATF6f/XBP1s (labeled in *Figure 7A* and *Figure 7—figure supplement 1A*). These sites are on the surface of the head domain (*Figure 8A*), in a region where positively charged residues, such as K170 and K176, increase host receptor binding and hinder antibody escape (*Hensley et al., 2009*). Additionally, the P175 site was the site most affected by XBP1s and ATF6f/XBP1s activation (*Figure 7A* and *Figure 7—figure supplement 1A*). Although P175 is a trans-proline in an unstructured coil, and is therefore less impactful than if found in a region of defined HA secondary structure, its mutational tolerance upon XBP1s induction could still be mediated by prolyl isomerase activity regulated by the UPR (*Shoulders et al., 2013*). Furthermore, several sites in the stem domain were strongly impacted by host ER proteostasis factors and temperature. For instance, mutational tolerance at the N497 and T499 sites was considerably reduced upon XBP1s induction, ATF6f/XBP1s induction, and upon increased temperature (*Figure 8* and *Figure 8—figure supplement 1*), likely because these residues constitute an N-linked glycosylation sequon. N-Linked glycosylation directs the co- and post-translational folding pathways of HA.

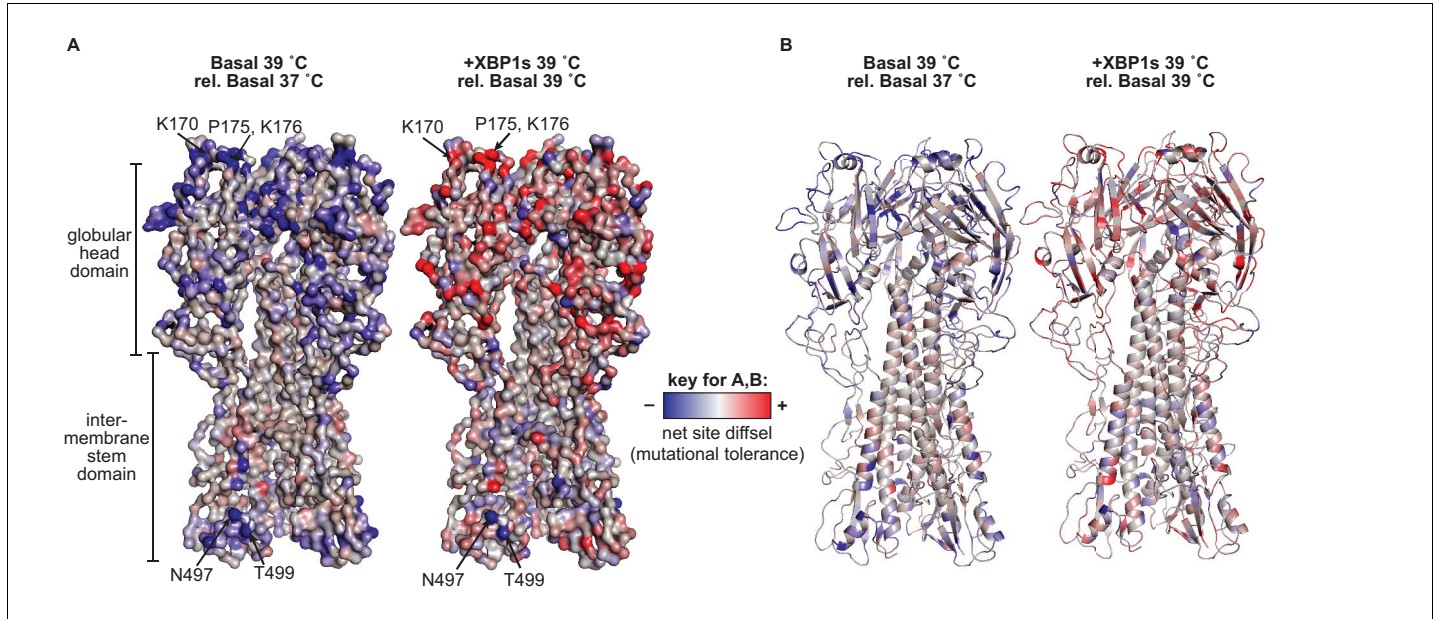

**Figure 8.** Mutational tolerance at sites across the mature HA protein was impacted by XBP1s induction and increased temperature. (**A**) Average net site diffsel values (which correspond to mutational tolerance) are mapped onto the HA crystal structure PDBID 1RVX (*Gamblin et al., 2004*), with select outliers from *Figure 7* labeled. (**B**) Cartoon representation of A displaying secondary structural elements. In A–B, negative net site diffsel values are mapped in blue; positive net site diffsel values are mapped in red. *Figure 8—figure supplement 1*. Mutational tolerance at sites across the mature HA protein was impacted by ATF6f/XBP1s induction and increased temperature.

DOI: https://doi.org/10.7554/eLife.38795.026

The following figure supplement is available for figure 8:

**Figure supplement 1.** Mutational tolerance at sites across the mature HA protein was impacted by ATF6f/XBP1s induction and increased temperature.
DOI: https://doi.org/10.7554/eLife.38795.027

Removal of this glycosylation sequon may reduce the interaction of HA with the glycoprotein chaperones calnexin and calreticulin, thereby hindering HA folding and trafficking (*Daniels et al., 2003*; *Hebert et al., 1997*). Altogether, host ER proteostasis factors and temperature impact the mutational tolerance at sites across the entire HA structure, irrespective of secondary structure and surface accessibility. Sites that are especially affected contain wild-type residues that likely mediate interactions between HA and host ER proteostasis factors.

## Discussion

Our results demonstrate that host proteostasis capacity and host cell temperature critically impact mutational tolerance across the entire HA protein. Activation of the XBP1s arm of the UPR globally enhanced the mutational tolerance of HA, while increased host cell temperature globally reduced the mutational tolerance of HA. Notably, the variants that were most enriched upon XBP1s induction corresponded to those most depleted upon increased temperature. This anti-correlation indicates that temperature-sensitive variants, which are likely biophysically deleterious relative to wild-type HA, are most likely to be rescued by the ER proteostasis machinery.

These selection pressures exerted similar fitness effects across the entire HA protein, influencing sites that differ in their secondary structure, surface accessibility, conservation in natural H1 sequences, and functional and structural roles. This impact is in stark contrast to selection pressures such as immune escape or drug resistance, which typically select for variants at a few specific sites. Universally modulating HA mutational tolerance suggests that host temperature and ER proteostasis mechanisms can shift the distribution of fitness effects of mutations on HA, and therefore are likely to broadly influence the accessibility of HA evolutionary trajectories. Because we observe a particularly strong impact at antigenic sites in the HA head domain (*Doud and Bloom, 2016*), inhibition of ER proteostasis mechanisms may limit the accessibility of antibody escape variants. This therapeutic

strategy is especially appealing because prior work has demonstrated that influenza infection can induce the unfolded protein response and that ER proteostasis mechanisms are essential for influenza propagation (*Frabutt et al., 2018*; *Hassan et al., 2012*).

Similar to previous studies on the role of HSP90 in the evolution of endogenous protein clients (*Cowen and Lindquist, 2005*), our observations may derive from direct interactions between host proteostasis components and HA, or as an indirect consequence of perturbing proteostasis. The negative correlation we observe between upregulating ER proteostasis factors and increasing temperature evidences the former, because variants with compromised fitness in a biophysically challenging environment are most impacted by the ER proteostasis machinery. Alternatively, ER proteostasis factors and increased temperature may affect the kinetics of infection or expression of third-party mediators that regulate HA function or fitness.

Irrespective of the precise mechanism, this study provides the first experimental evidence that ER proteostasis factors define the mutational tolerance of a secretory pathway client protein, a phenomenon that likely extends far beyond HA and viral evolution. Prior to this work, chaperones known to potentiate and buffer protein evolution were limited to cytosolic isoforms (*Phillips et al., 2017*), largely of HSP90 (*Rohner et al., 2013*; *Queitsch et al., 2002*; *Cowen and Lindquist, 2005*; *Geller et al., 2018*; *Sangster et al., 2008*; *Geiler-Samerotte et al., 2016*; *Geller et al., 2007*). Here, we show that this role extends to ER chaperones and quality control machinery involved in the folding and assembly of membrane and secretory proteins. We note that understanding the evolutionary constraints of membrane and secretory proteins is critical, as they comprise highly significant therapeutic targets, including collagen (*DiChiara et al., 2016*), G-protein coupled receptors (*Laschet et al., 2018*), antibodies (*Wong et al., 2018*), and viral membrane proteins (*Hurtley et al., 1989*). This work suggests that ER proteostasis mechanisms could be tuned to ameliorate protein-misfolding diseases or prevent the development of antiviral resistance. Further investigation into the impact of ER proteostasis factors on the mutational tolerance of additional secretory pathway client proteins, as well as non-client proteins, will elucidate the underlying molecular details and the pervasiveness of our findings.

# Materials and methods

**Key resources table**

| Reagent type (species) or resource | Designation | Source or reference | Additional information |
|---|---|---|---|
| strain, strain background (Influenza A virus) | Influenza A/WSN/1933 | PMID 27271655 | |
| genetic reagent (Influenza A virus) | Influenza A/WSN/1933 Mutant Library 1 | PMID 27271655 | |
| genetic reagent (Influenza A virus) | Influenza A/WSN/1933 Mutant Library 2 | PMID 27271655 | |
| genetic reagent (Influenza A virus) | Influenza A/WSN/1933 Mutant Library 3 | PMID 27271655 | |
| cell line (*Homo sapiens*) | HEK293$^{ATF6f/XBP1s}$ | PMID 23583182 | |
| recombinant DNA reagent | pHW181-PB2 | PMID 10801978 | plasmid |
| recombinant DNA reagent | pHW182-PB1 | PMID 10801978 | plasmid |
| recombinant DNA reagent | pHW183-PA | PMID 10801978 | plasmid |
| recombinant DNA reagent | pHW184-HA | PMID 10801978 | plasmid |
| recombinant DNA reagent | pHW185-NP | PMID 10801978 | plasmid |

*Continued on next page*

*Continued*

| Reagent type (species) or resource | Designation | Source or reference | Additional information |
|---|---|---|---|
| recombinant DNA reagent | pHW186-NA | PMID 10801978 | plasmid |
| recombinant DNA reagent | pHW187-M | PMID 10801978 | plasmid |
| recombinant DNA reagent | pHW188-NS1 | PMID 10801978 | plasmid |
| software, algorithm | dms_tools | PMID 25990960 | |

## Plasmids

The following plasmids were used for generating A/WSN/1933 influenza virus: pHW181-PB2, pHW182-PB1, pHW183-PA, pHW184-HA, pHW185-NP, pHW186-NA, pHW187-M, and pHW188-NS1 (*Hoffmann et al., 2000*).

## Cell culture

HEK293$^{ATF6f/XBP1s}$ cells were cultured at 37°C in a 5% $CO_2$ atmosphere in DMEM (CellGro) supplemented with 10% fetal bovine serum (CellGro) and 1% penicillin/streptomycin/glutamine (CellGro). Cells were generated from parental HEK293 cells (ATCC CRL-1573; authenticated by STR-profiling) as previously described (*Shoulders et al., 2013*). Here, 1 µM TMP and 0.1 µg/mL doxycycline were used to activate ATF6 and XBP1s, respectively. All cell lines were periodically tested for mycoplasma using the MycoSensor PCR Assay Kit from Agilent (302109).

## Influenza virus

All experiments were performed with influenza A/WSN/1933 (H1N1) HA viral mutant libraries (*Doud and Bloom, 2016*). All infections were performed in WSN media (OptiMEM-I; Thermo Fisher Scientific) supplemented with 0.5% heat-inactivated FBS, 0.3% BSA (Invitrogen), 100 U/mL of penicillin and 100 µg/mL of streptomycin (Bio Whittaker), and 100 µg/mL of $CaCl_2$); swapping the inoculum with fresh WSN media 2 hr post-infection.

## qPCR

HEK293$^{ATF6f/XBP1s}$ cells were seeded at 750,000 cells/well in a 12-well plate and treated with 0.01% DMSO, 0.1 µg/mL doxycycline (Sigma), or 0.1 µg/mL doxycycline and 1 µM TMP (Alfa Aesar) for 24 hr, or 100 µM arsenite for 2 hr as a positive control for heat shock response activation or 10 µg/mL tunicamycin for 6 hr as a positive control for unfolded protein response activation. To monitor chaperone levels during influenza infection, HEK293$^{ATF6f/XBP1s}$ cells were infected with influenza A/WSN/1933 at an MOI of 1 for 8 hr to mimic the environment of a cell infected with a single influenza virion as in the DMS. Cellular RNA was harvested using the Omega RNA Extraction kit with Homogenizer Columns. 1 µg RNA was used to prepare cDNA using random primers (total reaction volume = 20 µL; Applied Biosystems High-Capacity Reverse Transcription kit). The reverse transcription reaction was diluted to 80 µL with water, and 2 µL of each sample was used for qPCR with 2 × Sybr Green (Roche) and primers for human *RPLP2* (housekeeping gene), *HSP70*, *HSP40*, *HSP90*, *ERDJ4*, *GRP94*, *BIP*, *SEC24D*, and influenza *Matrix* (primer sequences in *Supplementary file 1*). All gene transcript levels were normalized to that of *RPLP2*, and the fold-change in expression relative to DMSO-treated, mock-infected cells was calculated (*Figure 2—figure supplement 3C*; *Figure 2—source data 3*). For qPCR of influenza-infected cells, a standard curve was prepared with a pDZ plasmid backbone containing the Influenza PR8 M segment to determine influenza *Matrix* copy number, which was used as a positive control for productive infection (*Figure 2—source data 3*).

## RNA-seq

For the HEK293$^{ATF6f/XBP1s}$ cell line characterization, HEK293$^{ATF6f/XBP1s}$ cells were seeded at 750,000 cells/well in a 12-well plate and treated with 0.01% DMSO, 0.1 µg/mL doxycycline, or 0.1 µg/mL doxycycline and 1 µM TMP for 24 hr, at either 37°C or 39°C. Cellular RNA was harvested using

Qiagen RNeasy Plus Mini Kit with QIAshredder homogenization columns. RNA-Seq libraries were prepared using the Kapa mRNA HyperPrep RNA-seq library construction kit system and were sequenced on an Illumina HiSeq SE40.

## RNA-seq analysis

Analyses were performed using tools and methods presented in *Huang et al. (2009)*. Quality control: Reads were aligned against hg19 (Feb., 2009) using bwa mem v. 0.7.12-r1039 [RRID:SCR_010910] with flags –t 16 –f and mapping rates, fraction of multiply-mapping reads, number of unique 20-mers at the 5′ end of the reads, insert size distributions and fraction of ribosomal RNAs were calculated using bedtools v. 2.25.0 [RRID:SCR_006646] (*Quinlan and Hall, 2010*). In addition, each resulting bam file was randomly down-sampled to a million reads, which were aligned against hg19 and read density across genomic features were estimated for RNA-Seq-specific quality control metrics.

RNA-Seq mapping and quantitation: Reads were aligned against GRCh38/ENSEMBL 89 annotation using STAR v. 2.5.3a with the following flags -runThreadN 8 –runMode alignReads –outFilterType BySJout –outFilterMultimapNmax 20 –alignSJoverhangMin 8 –alignSJDBoverhangMin 1 – outFilterMismatchNmax 999 –alignIntronMin 10 –alignIntronMax 1000000 –alignMatesGapMax 1000000 –outSAMtype BAM SortedByCoordinate –quantMode TranscriptomeSAM with –genomeDir pointing to a 75nt-junction GRCh38 STAR suffix array (*Dobin et al., 2013*). Gene expression was quantitated using RSEM v. 1.3.0 [RRID:SCR_013027] with the following flags for all libraries: rsem-calculate-expression –calc-pme –alignments -p 8 –forward-prob 0 against an annotation matching the STAR SA reference (*Li and Dewey, 2011*). Posterior mean estimates (pme) of counts and estimated RPKM were retrieved.

Differential expression analysis: Treatments were compared against DMSO-treatment (Basal) either at 37°C or 39°C for each condition (*Figure 2*; *Figure 2—figure supplement 1*; *Figure 2—source data 1*). Briefly, differential expression was performed in the R statistical environment (R v. 3.4.0) using Bioconductor's DESeq two package on the protein-coding genes only [RRID:SCR_000154] (*Love et al., 2014*). Dataset parameters were estimated using the estimateSizeFactors(), and estimateDispersions() functions; read counts across conditions were modeled based on a negative binomial distribution, and a Wald test was used to test for differential expression (nbinomWaldtest(), all packaged into the DESeq() function), using the treatment type as a contrast. Fold-changes and *p*-values were reported for each protein-coding gene. Heat maps were generated in Spotfire (Tibco), using a complete linkage function and the cosine correlation as distance metrics (*Figure 2*).

Comparison of upregulated genes in present study to previous study (*Shoulders et al., 2013*): This analysis was performed to compare genes upregulated upon XBP1s and ATF6f/XBP1s induction at 37°C, relative to DMSO treatment at 37°C. The analysis was restricted to genes with clear representation in both the whole-genome array (*Shoulders et al., 2013*) and the RNA-Seq annotation (n = 13058). Transcripts detected by the two methods were matched based on an Affymetrix probe ID for the whole-genome array and an ENSEMBL ID based on DAVID v6.8 for the RNA-Seq (*Huang et al., 2009*). Residual unmatched transcripts were matched based on NCBI IDs, and finally based on gene names (correcting for updates in official gene symbols between annotations). Both datasets were filtered applying a 2-fold change cutoff and an FDR of 0.05. The resulting overlap is presented in *Figure 2—figure supplement 2*. The genes for each subset of the Venn Diagram (*Figure 2—figure supplement 2*) are listed in *Figure 2—source data 2*. For each subgroup, ENSEMBL identities were retrieved for each gene and used in the DAVID online Gene Ontology environment (*Huang et al., 2009*) to call enriched functional categories against the background of 13058 genes jointly called across both RNA quantification methods (*Figure 2—source data 2*).

## Cell growth assay

HEK293^ATF6f/XBP1s cells were seeded at 750,000 cells/well in poly-D-lysine (Sigma)-coated (0.05 mg/ mL in PBS, 37°C for 15 min) 12-well plates and pre-treated with 0.01% DMSO, 0.1 μg/mL doxycycline, or 0.1 μg/mL doxycycline and 1 μM TMP, and placed at either 37°C or 39°C. To mimic the infection conditions, the cellular growth media was replaced with WSN media supplemented with 0.01% DMSO, 0.1 μg/mL doxycycline, or 0.1 μg/mL doxycycline and 1 μM TMP 16 hr post-treatment, returning plates to either 37°C or 39°C. 48 hr after the mock infection, the media was replaced

with WSN media containing 50 μM resazurin sodium salt (Sigma). After 4 hr of incubation, 100 μL of media was used to quantify resorufin fluorescence (excitation 530 nm; emission 590 nm) using a Take-3 plate reader (BioTeK) (*Figure 2—figure supplement 3A*).

## Influenza infection in modified host cell environments

HEK293[ATF6f/XBP1s] cells were seeded at 750,000 cells/well in poly-D-lysine-coated 12-well plates and treated with 0.01% DMSO, 0.1 μg/mL doxycycline, or 0.1 μg/mL doxycycline and 1 μM TMP, at either 37°C or 39°C, for 16 hr. To assess viral growth in each selection condition, pre-treated cells were infected with influenza A/WSN/1933 at an MOI of 0.01 virions/cell for 48 hr, at either 37°C or 39°C, to mimic DMS and pairwise competition conditions. Viral supernatant was harvested 48 hr post-infection and titered using a $TCID_{50}$ assay (*Figure 2—figure supplement 3B*).

## Deep mutational scanning

We employed three biological replicate viral libraries that were previously generated from three independently prepared HA mutant plasmid libraries (*Doud and Bloom, 2016*). HEK293[ATF6f/XBP1s] cells were plated in poly-D-lysine-coated 15 cm plates at a density of $50 \times 10^6$ cells per dish; 8 hr after plating, cells were treated with 0.01% DMSO, 0.1 μg/mL doxycycline, or 0.1 μg/mL doxycycline and 1 μM TMP for 16 hr, at either 37°C or 39°C. 16 hr after treatment, $1 \times 10^6$ Infectious virions (determined using a $TCID_{50}$ assay) from each viral library were used to infect two 15 cm plates in each condition at an MOI of 0.01 virions/cell. In addition, one 15 cm plate at both 37°C and 39°C was either mock infected (negative control) or infected with wild-type virus. For infection, the cellular growth medium was replaced with WSN media containing a mutant virus library, wild-type virus, or no virus for mock infection. After 2 hr, the inoculum was replaced with fresh WSN media containing 0.01% DMSO, 0.1 μg/mL doxycycline, or 0.1 μg/mL doxycycline and 1 μM TMP. 48 hr post-infection, the viral supernatant was harvested, centrifuged at $1000 \times g$ for 5 min to remove cell debris, and stored at –80°C. To extract viral RNA from the viral supernatant, the supernatant was thawed and virions were concentrated by ultracentrifugation, spinning at 25,000 rpm for 2 hr at 4°C (SW 32 Ti swinging bucket rotor, Beckman Coulter). The supernatant was then decanted, and the virion pellet was resuspended in 140 μL QIAgen Viral RNA Buffer AVL. Viral RNA was then extracted using a QIAgen Viral RNA mini kit, per the manufacturer's instructions, changing collection tubes at each step. Viral RNA was reverse transcribed using the AccuScript High Fidelity first strand cDNA synthesis kit (Agilent) using 5'-WSN-HA and 3'-WSN-HA primers (primer sequences in *Supplementary file 1*). At least $10^6$ HA molecules were PCR-amplified for preparation of subamplicon sequencing libraries, as previously described (*Doud and Bloom, 2016*), to ensure sufficient sampling of viral library diversity. Briefly, this sequencing library preparation method appends unique, random barcodes and part of the Illumina adapter to HA subamplicon molecules. In a second round of PCR, the complexity of the uniquely barcoded subamplicons was controlled to be less than the sequencing depth, and the remainder of the Illumina adapter is appended (*Doud and Bloom, 2016*). The resulting libraries were sequenced on an Illumina HiSeq 2500 in rapid run mode with $2 \times 250$ bp paired-end reads.

## Deep mutational scanning data analysis

The software dms_tools (http://jbloomlab.github.io/dms_tools/) (*Bloom, 2015*) was used to align reads to the Influenza A/WSN/1933 HA reference sequence, count amino acid variants across HA, and calculate the differential selection for each variant between two selection conditions, as previously described (*Doud et al., 2017*). Briefly, reads were trimmed, aligned to the Influenza A/WSN/1933 reference sequence, and read pairs were quality filtered by discarding read pairs with low-quality sites in the barcode region of either read, in addition to read pairs with more than 7.5% low quality sites. The remaining read pairs were sorted by barcode and barcodes with less than two read pairs were discarded. For each remaining barcode, mutation calls were only made when greater than 75% of reads concurred (*Doud and Bloom, 2016*; *Bloom, 2015*). For each sample, the numbers of read pairs, barcoded read pairs, and reads per barcode are reported in *Source data 1* (see sample _summarystats.txt) and the sequencing data analyses are available at https://github.com/amphilli/HA_DMS_2018 (copy archived at https://github.com/elifesciences-publications/amphilli/HA_DMS_2018). For evaluating mutation differential selection, variants not present in each replicate starting library were removed from the analysis and were subsequently filtered for variants that

incurred selection of the same sign across biological triplicates (*Figure 5—source data 1*). The mutation differential selection values for these filtered variants were visualized on sequence logo plots, made using dms_tools (*Figure 4*; *Figure 4—figure supplements 1–5*) (*Doud et al., 2017*; *Bloom, 2015*). Correlation between mutation differential selection values across selection conditions was determined using linear regression using Graph Pad Prism software, reporting Pearson correlation coefficients (*Figure 5B–C*; *Figure 5—figure supplement 1*). For each site in HA, the net site differential selection (net site diffsel) was calculated by summing the mutation differential selection values separately for each replicate starting library, excluding mutation differential selection values for variants that were not present in the respective starting library. These replicate net site diffsel values were then then averaged and sorted by HA secondary structure, relative surface accessibility (*Gamblin et al., 2004*; *Joosten et al., 2011*), receptor binding and antigenic sites (*Doud and Bloom, 2016*), N-glycosylation sequons and disulfide bond sites (*Chen et al., 1995*; *Daniels et al., 2003*; *Hebert et al., 1997*), and natural HA site conservation (*Zhang et al., 2017*) (*Figure 7*; *Figure 7—figure supplement 1*; *Figure 7—source data 1*). HA secondary structure and relative surface accessibility were determined using DSSP (*Joosten et al., 2011*; *Kabsch and Sander, 1983*) based on the HA crystal structure (PDBID 1RVX) (*Gamblin et al., 2004*). In *Figure 7A*, sites designated buried were the 50% least surface-accessible sites; sites designated exposed were the 50% most surface-accessible sites. Receptor binding and antigenic sites were classified as in Doud *et al*, and included sites nearby antigenic sites (*Doud and Bloom, 2016*). Natural HA site conservation was based on 989 non-redundant human H1 sequences preceding the 2009 H1N1 pandemic, accessed from the NIAID Influenza Research Database (accessed Jan. 10[th], 2018) (*Zhang et al., 2017*). Site entropies were calculating using the Influenza Research Database SNP Analysis Tool (*Zhang et al., 2017*), and were normalized to a scale of 0–100, with a site entropy of 100 representing the most variable site in HA and 0 representing the most conserved site (*Figure 7B*). In *Figure 7A*, sites designated conserved had a site entropy of zero; all other sites were designated as variable. The net site differential selection values were mapped onto the HA crystal structure (PDBID 1RVX) (*Gamblin et al., 2004*) using PyMOL (*DeLano, 2002*) (*Figure 8*; *Figure 8—figure supplement 1*).

## Data availability

FASTQ files for DMS sequencing are available in the Sequence Read Archive under accession number SRP149672. The deep mutational scanning data analyses are available at https://github.com/amphilli/HA_DMS_2018, and is also available in *Source data 1*. All mutation differential selection values from deep mutational scanning (pre- and post-filtering) are available in *Figure 5—source data 1*. The complete RNAseq data are available from GEO under accession number GSE115168.

## Infectious viral titering via tissue culture infectious dose (TCID$_{50}$) assay

10-Fold dilutions of each virus were prepared in quadruplicate in 96-well plates. 5,000 MDCK-SIAT1 cells were then added to each well and incubated at 37°C for 72 hr, after which the wells were scored for the presence of cytopathic effect. The dilutions of virus displaying cytopathic effect in the MDCK-SIAT1 cells were then used to calculate the TCID$_{50}$/μL using https://github.com/jbloomlab/reedmuenchcalculator as described by Thyagarajan and Bloom (*Thyagarajan and Bloom, 2014*), where virions/μL = 0.69*TCID$_{50}$/μL.

## Reverse genetics pairwise viral competitions

HA mutant-encoding plasmids were prepared by introducing point mutations into the pHW184-HA plasmid using the QuikChange II XL Site-Directed Mutagenesis Kit (Agilent 200522). Mutant viruses were generated from plasmids by transfecting a co-culture of $2.5 \times 10^4$ MDCK-SIAT1 and $3 \times 10^5$ HEK 293 T cells, as previously described (*Hoffmann et al., 2000*), followed by titering using the TCID$_{50}$ assay. For each competition, HEK293$^{ATF6f/XBP1s}$ cells were plated in poly-D-lysine-coated 24-well dishes at $3.5 \times 10^5$ cells/well and treated 8 hr later with 0.01% DMSO, 0.1 μg/mL doxycycline, or 0.1 μg/mL doxycycline and 1 μM TMP at either 37°C or 39°C. 24 hr after plating, cells were infected with a 1:1 mixture of wild-type and mutant viruses at an MOI of 0.01 virions/cell in biological triplicate under conditions identical to that of the deep mutational scanning experiment. After 2 hr, the inoculum was replaced with fresh WSN media supplied with 0.01% DMSO, 0.1 μg/mL doxycycline, or 0.1 μg/mL doxycycline and 1 μM TMP. 48 hr post-infection, infectious supernatant was

harvested, centrifuged at 1000 × g for 5 min to remove cell debris, and stored at –80°C. Viral RNA was extracted from 140 μL infectious supernatant using the QIAamp Viral RNA Mini kit and at least $10^6$ HA molecules were reverse transcribed using SuperScript III Reverse Transcriptase (Thermo Fisher Scientific) with 5'-WSN-HA and 3'-WSN-HA primers (*Supplementary file 1*). The dsDNA was purified twice using 0.9 × AMPure XP beads (Beckman Coulter) and quantified using a Quant-iT PicoGreen assay (Life Technologies). Illumina NexteraXT sequencing libraries were prepared using a Mosquito HTS liquid handler (TTP Labtech) and sequenced on an Illumina HiSeq with 50 bp single-end reads.

### Pairwise competition sequencing data analysis

To overcome biases in read mapping from consecutive, multi-nucleotide sequence variants, mapping target sequences were generated for each Influenza A/WSN/1933 HA variant. Briefly, regions flanking the variant (given the experimental read length) would be identified, relevant groups of bases substituted with their engineered variants, and the rest of the sequence substituted with 'N' in order to maintain the sequence position of said variants, thus generating sets of 'pseudo-haplotypes'. Sequencing reads were aligned to the wild-type Influenza A/WSN/1933 HA sequence and the set of pseudo-haplotypes using bwa mem (v. 0.7.12-r1039) (arXiv:1303.3997), with flag –t 16, and sorted and indexed bam files were generated using samtools (v 1.3) (*Li et al., 2009*). These bam files were processed using samtools mpileup with flags –excl-flags 2052, -d 30000000 and the same reference sequences used for mapping (*Sievers et al., 2011*). Allele counts at each position of the wild-type and pseudo-haplotype sequences were summed and reported along the wild-type Influenza A/WSN/1933 HA sequence for each sample for calculation of variant frequencies. The change in variant frequency upon selection was normalized to that of wild-type, thus providing diffsel values on the same scale as the DMS competition (*Figure 6B*; *Figure 6—figure supplement 1*).

### Statistics

All experiments were performed in biological triplicate with replicates defined as independent experimental entireties (i.e., from plating the cells to acquiring the data). For deep mutational scanning, each biological replicate mutant viral library was prepared from independently generated mutant plasmid libraries, as previously reported (*Doud and Bloom, 2016*). Differential selection values from deep mutational scanning, as well as selection values from pairwise competitions, were tested for significance of deviation from zero (wild-type behavior), using a one-sample *t*-test in Graph Pad Prism (*Figure 5A*; *Figure 6—figure supplement 1*). Differential selection values from pairwise competitions were compared between selection conditions by a Student's *t*-test (*Figure 6—figure supplement 1*). Correlation between diffsel values was determined by linear regression using Graph Pad Prism software, reporting Pearson correlation coefficients and *p*-values from *F*-tests, testing significance of slope deviating from zero (*Figure 5B–C*; *Figure 5—figure supplement 1*). Correlation between site diffsel values (N = 565) was determined by linear regression using Pandas Python package, reporting Pearson correlation coefficients (*Figure 3—source data 1*). For DMS mutational tolerance analyses, statistical significance of the deviation of the mean of the net site diffsel distribution from zero (wild-type behavior) was tested by a Student's *t*-test; one-tailed *p*-values are reported, to assess whether the net site diffsel distributions for XBP1s and ATF6f/XBP1s induction are significantly greater than zero, and whether the net site diffsel distribution for increased temperature is significantly less than zero (*Figure 7A*; *Figure 7—figure supplement 1*; *Figure 7—source data 1*).

## Acknowledgements

The authors would like to thank Christopher E R Richardson (MIT), Andrew S DiChiara (AbbVie), and Chet M Berman (MIT) for feedback on this manuscript.

## Additional information

### Funding

| Funder | Grant reference number | Author |
|---|---|---|
| National Science Foundation | Graduate Research Fellowship | Angela M Phillips |
| National Cancer Institute | Koch Institute Support (core) Grant P30-CA14051 | Vincent L Butty Matthew D Shoulders |
| National Institute of Environmental Health Sciences | MIT Center for Environmental Health Sciences (core) Grant P30-ES002109 | Vincent L Butty Matthew D Shoulders |
| Tufts University | | Yu-Shan Lin |
| National Institutes of Health | AI127893 | Jesse D Bloom |
| Howard Hughes Medical Institute | Faculty Scholars grant | Jesse D Bloom |
| Simons Foundation | Faculty Scholars grant | Jesse D Bloom |
| National Science Foundation | CAREER Award 1652390 | Matthew D Shoulders |
| Richard and Susan Smith Family Foundation | Award for Excellence in Biomedical Research | Matthew D Shoulders |
| Massachusetts Institute of Technology | | Matthew D Shoulders |
| National Institutes of Health | 1DP2GM119162 | Matthew D Shoulders |

The funders had no role in study design, data collection and interpretation, or the decision to submit the work for publication.

### Author contributions

Angela M Phillips, Conceptualization, Formal analysis, Funding acquisition, Investigation, Visualization, Writing—original draft, Writing—review and editing; Michael B Doud, Resources, Methodology, Writing—review and editing; Luna O Gonzalez, Formal analysis, Investigation, Writing—review and editing; Vincent L Butty, Data curation, Formal analysis, Writing—review and editing; Yu-Shan Lin, Resources, Formal analysis, Visualization, Writing—review and editing; Jesse D Bloom, Resources, Software, Supervision, Methodology, Writing—review and editing; Matthew D Shoulders, Conceptualization, Supervision, Funding acquisition, Project administration, Writing—review and editing

### Author ORCIDs

Angela M Phillips http://orcid.org/0000-0002-9806-7574
Michael B Doud http://orcid.org/0000-0002-8172-6342
Yu-Shan Lin http://orcid.org/0000-0001-6460-2877
Jesse D Bloom http://orcid.org/0000-0003-1267-3408
Matthew D Shoulders http://orcid.org/0000-0002-6511-3431

### Decision letter and Author response

Decision letter https://doi.org/10.7554/eLife.38795.036
Author response https://doi.org/10.7554/eLife.38795.037

## Additional files

### Supplementary files

• Source data 1. Complete analysis of deep mutational scanning data.
DOI: https://doi.org/10.7554/eLife.38795.028
• Supplementary file 1. Primer sequences for qPCR and HA sequencing.

DOI: https://doi.org/10.7554/eLife.38795.029

• Transparent reporting form
DOI: https://doi.org/10.7554/eLife.38795.030

### Data availability

FASTQ files for DMS sequencing are available in the Sequence Read Archive under accession number SRP149672. The deep mutational scanning data analysis will be available upon publication at https://github.com/amphilli/HA_DMS_2018 (copy archived at https://github.com/elifesciences-publications/amphilli/HA_DMS_2018), and is also available in Source Data 1. All differential selection values from deep mutational scanning (pre- and post-filtering) are available in Figure 5-source data 1. The complete RNAseq data are available from GEO under accession number GSE115168.

The following datasets were generated:

| Author(s) | Year | Dataset title | Dataset URL | Database, license, and accessibility information |
|---|---|---|---|---|
| Angela M Phillips, Michael B Doud, Luna O Gonzalez, Vincent L Butty, Yu-Shan Lin, Jesse D Bloom, Matthew D Shoulders | 2018 | ER proteostasis and temperature differentially impact the mutational tolerance of influenza hemagglutinin | https://www.ncbi.nlm.nih.gov/sra/SRP149672 | Publicly available at the NCBI Gene Expression Omnibus (accession no: SRP149672) |
| Angela M Phillips, Michael B Doud, Luna O Gonzalez, Vincent L Butty, Yu-Shan Lin, Jesse D Bloom, Matthew D Shoulders | 2018 | ER proteostasis and temperature differentially impact the mutational tolerance of influenza hemagglutinin | https://www.ncbi.nlm.nih.gov/geo/query/acc.cgi?acc=GSE115168 | Publicly available at the NCBI Gene Expression Omnibus (accession no: GSE115168) |

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
