## [Decision Letter]

Thank you for submitting your article "Enhanced ER proteostasis and temperature differentially impact the mutational tolerance of influenza hemagglutinin" for consideration by *eLife*. Your article has been reviewed by three peer reviewers, including Ulrich Hartl as the Reviewing Editor and Reviewer #1, and the evaluation has been overseen by Wenhui Li as the Senior Editor. The following individuals involved in review of your submission have agreed to reveal their identity: Raul Andino (Reviewer #2); Jeffery W Kelly (Reviewer #3).

The reviewers have discussed the reviews with one another and the Reviewing Editor has drafted this decision to help you prepare a revised submission.

Summary:

In this manuscript, Philips et al. explore the impact of ER proteostasis on the mutational and fitness landscape of a model substrate of the secretory pathway – influenza virus hemagglutinin (HA). The authors extend upon their (and others') previous work on host cell cytosolic proteostasis influence on viral sequence space.

The authors make use of a saturating library of single site HA mutants in a cellular assay of infection and replicative fitness of the mutant virions in comparison to the wild type HA in various conditions. Using targeted deep sequencing of HA, they were able to identify the spectrum of mutations which were either enriched or depleted (deep mutational scanning or DMS) under these conditions. They show that enhancing host cell ER proteostasis broadly increases mutational tolerance across the length of the model substrate HA and is likely to be relevant to shaping evolutionary trajectories of endogenous secretory pathway proteins. Importantly, they observe a striking rescue of high-temperature sensitive mutants by such an enhanced ER proteostatic state.

The data presented is comprehensive and extends our understanding of how compartment specific proteostasis mechanisms assist in shaping protein evolution.

Essential revisions:

1) A major issue with the authors’ central conclusion is that they did not examine other viral protein. If the proteostasis machinery has a direct effect on HA evolution it would be important to present evidence that other viral proteins are not affected, e.g. those that are not translocated into the ER. This point is important because it is possible that upregulation of the ER proteostasis pathway may result in pleiotropic and indirect effects on virus evolution. For example, ER proteostasis factors may affect expression of other factors that regulate HA function. Indeed, XBP1s regulates ~350 genes and ATF6f also affects a number of chaperones and quality control components. If other viral proteins are not analyzed, it is difficult to conclude that the observed effect is direct on client proteins.

It is suggested that the authors make a clear statement to the effect that their data cannot exclude the possibility that proteostasis effects not specific for the ER play a role.

2) Similarly, the increase on temperature (39°C) could modify factors and the kinetics of infection that indirectly can affect selection on HA variants. For example, have the authors considered the effect on translation and ER translocation rates?

This point could be addressed either experimentally or verbally.

3) In Figure 2—figure supplement 1, the authors examine the modified background conditions prior to conducting the DMS using RNAseq and claim consistency with their previous dataset. A visual comparison in the form of a Venn diagram (fold change based, at least, given that their previous dataset was from microarrays) can be presented to clarify the degree of overlap and outliers, if any.

4) Subsection “Modulating ER proteostasis during influenza infection”, third paragraph. The authors state that reduced growth did not affect the interpretation of the data. They should provide the average depth per sample and other related statistics either as a supplementary table or describe this in the methods. A replicate fidelity measure like a correlation coefficient or simple PCA would add to the validity of the dataset as a whole.

5) In the section describing the data filtering (subsection “Deep mutational scanning of HA in modulated ER proteostasis environments”, third paragraph), the reader would like to know the actual dropout percentages as per exclusion criteria (given numerically as a table).

6) In Figure 5A, the authors describe that at high temperature there is a gross reduction in the variant growth, consistent with the hypothesis of biophysical constraints imposed on them. Physiologically, this is likely due to a misfolding and degradation of the variants. Given that host cells do not upregulate components of ERAD at higher temperature (Figure 2 and Figure 2—figure supplement 1E) – is ERAD engaged more at such temperatures without burdening the overall proteostasis? Also, in conditions of UPR activation, at permissive and higher temperatures, the RNAseq data reveal highly upregulated ERAD components (in addition to other factors). Should the same variants not be subject to degradation? How many variants do not get enriched upon the ER proteostasis boost? The authors could explain their results in some more specific details.

7) Again related to Figure 5A, in the subsection “Deep mutational scanning of HA in modulated ER proteostasis environments”, fifth paragraph, describe the redundancy of the variants having a positive diffsel score between XBP1s and ATF6f/XBP1s conditions. Although the authors comment about the similarity, it would be interesting to see which sites overlap and which do not, possibly shedding light on subtle differences between the two modes of UPR activation.

8) Figure 7A describes the mutational tolerance in the XBP1s condition alone. What about the ATF6f/XBP1s condition?

9) In Figure 7B and 7C, the y-axis is unclear – is it a continuum from conserved to variable? Buried to exposed?

---

## [Author Response]

Essential revisions:1) A major issue with the authors’ central conclusion is that they did not examine other viral protein. If the proteostasis machinery has a direct effect on HA evolution it would be important to present evidence that other viral proteins are not affected, e.g. those that are not translocated into the ER. This point is important because it is possible that upregulation of the ER proteostasis pathway may result in pleiotropic and indirect effects on virus evolution. For example, ER proteostasis factors may affect expression of other factors that regulate HA function. Indeed, XBP1s regulates ~350 genes and ATF6f also affects a number of chaperones and quality control components. If other viral proteins are not analyzed, it is difficult to conclude that the observed effect is direct on client proteins.It is suggested that the authors make a clear statement to the effect that their data cannot exclude the possibility that proteostasis effects not specific for the ER play a role.

We agree with the reviewers that our data cannot exclude the possibility of secondary effects of ER proteostasis on HA mutational tolerance. In the original submission, we briefly addressed this possibility in the Discussion. We have now expanded that discussion to include the possible secondary effects of increased temperature (as requested in Essential revision #2, below), and also to explicitly state that future work is necessary to determine if the observed effects are specific to secretory pathway client proteins:

“Similar to previous studies on the role of HSP90 in the evolution of endogenous protein clients (Cowen and Lindquist, 2005), our observations may derive from direct interactions between host proteostasis components and HA, or as an indirect consequence of perturbing proteostasis. […] Alternatively, ER proteostasis factors and increased temperature may affect the kinetics of infection or expression of third-party mediators that regulate HA function or fitness.”

“Further investigation into the impact of ER proteostasis factors on the mutational tolerance of additional secretory pathway client proteins, as well as non-client proteins, will elucidate the underlying molecular details and the pervasiveness of our findings.”

2) Similarly, the increase on temperature (39°C) could modify factors and the kinetics of infection that indirectly can affect selection on HA variants. For example, have the authors considered the effect on translation and ER translocation rates?This point could be addressed either experimentally or verbally.

We have addressed this point verbally (see above statement from Discussion).

3) In Figure 2—figure supplement 1, the authors examine the modified background conditions prior to conducting the DMS using RNAseq and claim consistency with their previous dataset. A visual comparison in the form of a Venn diagram (fold change based, at least, given that their previous dataset was from microarrays) can be presented to clarify the degree of overlap and outliers, if any.

We now added the requested comparative analysis for the induction of XBP1s alone or with ATF6f specifically at 37˚C, as only this temperature was tested in our previous dataset (Shoulders et al., 2013). Per the request, this comparison is now presented as a Venn diagram in Figure 2—figure supplement 2. We also expanded our discussion of this dataset comparison to address the degree of overlap and possible explanations for discrepancies (e.g. different treatment conditions and transcriptomic measurements), as follows:

“The media composition and treatment regimen for influenza propagation differed from the conditions used in the original characterization of XBP1s and ATF6f/XBP1s activation in these cells, which also was performed only at 37 ˚C (Shoulders et al., 2013). […] Still, as expected (Shoulders et al., 2013), transcripts upregulated upon XBP1s and ATF6f/XBP1s induction largely corresponded to secretory pathway and ER stress response genes (Figure 2—figure supplement 1 and Figure 2—source data 2).”

4) Subsection “Modulating ER proteostasis during influenza infection”, third paragraph. The authors state that reduced growth did not affect the interpretation of the data. They should provide the average depth per sample and other related statistics either as a supplementary table or describe this in the methods. A replicate fidelity measure like a correlation coefficient or simple PCA would add to the validity of the dataset as a whole.

We agree with the reviewer that these sequencing statistics are important for assessing the validity of the dataset. In the original submission, the number of read pairs, number of barcoded read pairs, and number of reads per barcode were reported for each sample in Source Data 1. We now realize that this information was not easily accessible, and revised the Materials and Methods to explicitly describe the sequencing library preparation and data processing and also specified the precise location of this information in Source Data 1. We also note that the samples with reduced viral growth were held to identical library preparation and sequencing data analyses standards, such that the diversity of each mutant viral library was sufficiently sampled and accurately quantified. The revised discussion follows below:

“At least 10^6^ HA molecules were PCR-amplified for preparation of subamplicon sequencing libraries, as previously described (Russell, Trapnell and Bloom, 2018), to ensure sufficient sampling of viral library diversity. […] In a second round of PCR, the complexity of the uniquely barcoded subamplicons is controlled to be less than the sequencing depth, and the remainder of the Illumina adapter is appended (Russell, Trapnell and Bloom, 2018).”

“dms_tools (http://jbloomlab.github.io/dms_tools/) (Doud and Bloom, 2016) was used to align reads to the Influenza A/WSN/1933 HA reference sequence, count amino acid variants across HA, and calculate the differential selection for each variant between two selection conditions, as previously described (Doud, Hensley and Bloom, 2017). […] For each sample, the number of read pairs, barcoded read pairs, and reads per barcode are reported in Source Data 1 (see sample _summarystats.txt) and the complete sequencing data analyses are available at https://github.com/amphilli/HA_DMS_2018.”

Furthermore, per the reviewers’ suggestion, we evaluated the correlation between unfiltered site differential selection values for biological replicates, and have provided these correlation coefficients in Figure 3—source data 1. Additionally, we have added a statement to the Results to summarize the correlation between replicates:

“Notably, the unfiltered biological replicate diffsel values were strongly correlated across HA sites, with correlation coefficients ranging from *R* = 0.67–0.78 (Figure 3—source data 1 and Source Data 1).”

5) In the section describing the data filtering (subsection “Deep mutational scanning of HA in modulated ER proteostasis environments”, third paragraph), the reader would like to know the actual dropout percentages as per exclusion criteria (given numerically as a table).

There is now a table presented in Figure 3C summarizing the percentage of amino acid space sampled by each replicate mutant library, and also the percentage sampled for all three replicate libraries, and hence passing the filtering criteria. We note that, while calculating these numbers in revision, we noticed a mistake in our original filtering code that caused it to be falsely stringent for select phenylalanine variants and falsely relaxed for select glycine variants. After fixing this mistake and re-analyzing the data, the change in whether these variants passed our filters or not had no significant impact on the results or on any of our analyses.

6) In Figure 5A, the authors describe that at high temperature there is a gross reduction in the variant growth, consistent with the hypothesis of biophysical constraints imposed on them. Physiologically, this is likely due to a misfolding and degradation of the variants. Given that host cells do not upregulate components of ERAD at higher temperature (Figure 2 and Figure 2—figure supplement 1E) – is ERAD engaged more at such temperatures without burdening the overall proteostasis? Also, in conditions of UPR activation, at permissive and higher temperatures, the RNAseq data reveal highly upregulated ERAD components (in addition to other factors). Should the same variants not be subject to degradation? How many variants do not get enriched upon the ER proteostasis boost? The authors could explain their results in some more specific details.

Reduction in variant growth at 39˚C may be caused by enhanced ER-associated degradation (ERAD), as the reviewer suggests, or it may be caused by reduced variant folding rates (and thus slower variant viral growth). Though we do not observe upregulation of ERAD factors (Figure 2 and Figure 2—figure supplement 1E), our data do not speak to the activity of ERAD components at elevated temperature. As the reviewer suggests, it is certainly possible that ERAD components are engaged more at elevated temperature, resulting in the degradation of HA variants without inducing upregulation of ERAD factors.

Also as the reviewer suggests, UPR activation indeed upregulates ERAD components, in addition to ER chaperones and other proteostasis factors (Figure 2 and Figure 2—figure supplement 1A–D). Thus, variants could certainly be either assisted or degraded by induction of the UPR. The number of enriched and depleted variants is now noted in Figure 5A, though we emphasize that these numbers are all relative to how the wild-type HA performs, and thus positive and negative values do not reflect folding and degradation, respectively. We modified our Results section to elaborate on these points:

“This analysis confirmed that increased temperature indeed generally reduced mutant HA viral growth, resulting in depletion (i.e. diffsel < 0) of 597 variants versus enrichment (i.e. diffsel > 0) of only 361 variants, relative to wild-type HA (Figure 5A). […] For example, these perturbations could potentially impact the protein levels and activities of other host proteins that engage HA, or perhaps differentially modulate viral growth kinetics.”

7) Again related to Figure 5A, in the subsection “Deep mutational scanning of HA in modulated ER proteostasis environments”, fifth paragraph, describe the redundancy of the variants having a positive diffsel score between XBP1s and ATF6f/XBP1s conditions. Although the authors comment about the similarity, it would be interesting to see which sites overlap and which do not, possibly shedding light on subtle differences between the two modes of UPR activation.

We agree that this comparison is interesting and have added a correlation plot comparing activation of ATF6f/XBP1s and XBP1s as Figure 5—figure supplement 1. Outliers that are highly enriched or depleted in both conditions are labeled, and the full list of differential selection values for variants in these selection conditions is provided in Figure 5—source data 1. We note that few sites are strongly enriched in one condition and strongly depleted in the other. We also added references to Figure 5—figure supplement 1 and Figure 5—source data 1 to support our comments in the manuscript:

“The enhanced variant growth upon XBP1s induction alone was quite similar to that of simultaneous induction of XBP1s and ATF6f (Figure 5—figure supplement 1 and Figure 5—source data 1), and is thus either predominantly caused by XBP1s, or by XBP1s-regulated factors that are redundant with those regulated by ATF6f.”

8) Figure 7A describes the mutational tolerance in the XBP1s condition alone. What about the ATF6f/XBP1s condition?

Similar to the mutation differential selection distributions presented in Figure 5A, the site differential selection distributions (i.e. mutational tolerance distributions) for ATF6f/XBP1s are essentially identical to that of XBP1s alone, and were thus excluded from the original manuscript in an effort to simplify Figure 7. To address the reviewer’s question, we have now presented this analysis for ATF6f/XBP1s induction in Figure 7—figure supplement 1, and have commented on their similarity in the manuscript:

“We note that, similar to the mutation diffsel distributions (Figure 5A), the site diffsel distributions for ATF6f/XBP1s induction mirrored those for XBP1s induction (Figure 7—figure supplement 1).”

Furthermore, we also mapped the site diffsel values onto the HA crystal structure for ATF6f/XBP1s induction (Figure 8—figure supplement 1):

“Mapping net site diffsel values onto the HA crystal structure revealed that these selection pressures affect the mutational tolerance of sites across HA (Figure 8; Figure 8—figure supplement 1).”

9) In Figure 7B and 7C, the y-axis is unclear – is it a continuum from conserved to variable? Buried to exposed?

We agree that the y-axis in the original manuscript was unclear and have provided more detailed figure legends for Figures 7B–C to address this concern:

“(B) Net site diffsel is plotted as a function of site conservation, where x = 0 corresponds to a completely conserved site among 989 non-redundant human H1 sequences (1918–2008) (38). (C) Net site diffsel is plotted as a function of site surface accessibility, where x = 0 corresponds to a buried residue in the (PDBID 1RVX (32)).”